# Annual Censuses and Citizen Science Data Show Rapid Population Increases and Range Expansion of Invasive Rose-Ringed and Monk Parakeets in Seville, Spain

**DOI:** 10.3390/ani12060677

**Published:** 2022-03-08

**Authors:** Dailos Hernández-Brito, Martina Carrete, José L. Tella

**Affiliations:** 1Department of Conservation Biology, Doñana Biological Station (CSIC), Calle Américo Vespucio, 26, 41092 Sevilla, Spain; tella@ebd.csic.es; 2Department of Physical, Chemical and Natural Systems, Universidad Pablo de Olavide, Carretera de Utrera, km 1, 41013 Sevilla, Spain; mcarrete@upo.es

**Keywords:** biological invasions, *Psittacula krameri*, *Myiopsitta monachus*, population, distribution, citizen science, urban environment

## Abstract

**Simple Summary:**

Monitoring programs are crucial to understanding and managing invasive species populations. However, they are infrequent and not usually conducted in the long term. In this work, we used population censuses and observational data from citizen science platforms to monitor the growth and expansion of populations of two invasive species established in Seville (Spain): the rose-ringed parakeet and the monk parakeet. During our study period (2013–2021), rose-ringed and monk parakeet populations increased fivefold and twentyfold, respectively. These rapid population growths coincided with the increasing number of observations of both species recorded by volunteer birdwatchers, as well as the increasing expansion of monk parakeets throughout the study area. Citizen science can be useful for roughly knowing the population status of invasive species, but it cannot replace specific monitoring programs to understand their spatiotemporal dynamics.

**Abstract:**

Population changes of invasive species can go unnoticed long before population explosions, so long-term monitoring programs are needed to assess changes in population size. Although invasive populations of rose-ringed (*Psittacula krameri*) and monk parakeets (*Myiopsitta monachus*) are present worldwide, their current status and dynamics are mostly poorly known. Here, we provide a long-term population monitoring of both parakeet species established in a Mediterranean urban area. Between 2013 and 2021, we conducted systematic population censuses in the city of Seville and collected their occurrence and spatial distribution data from citizen science platforms. Our censuses showed a rapid population growth of both species: rose-ringed parakeets increased from 1200 to 6300 individuals, while monk parakeets increased from 70 to 1487 individuals. These population trends were weakly reflected by the number of parakeet observations and the number of cells with parakeet observations but not by the number of individuals recorded in citizen science platforms. Moreover, for the monk parakeet, the number of cells with observations was related to the spatial spread of its nests across the study area. Although resource-intensive, long-term monitoring programs are essential to assess population changes and develop effective management actions for invasive species. Thus, contrasting this information with data taken through citizen science platforms can validate the utility of the latter for assessing population status of invasive species.

## 1. Introduction

Invasive species are among the main drivers of global change due to their ecological impacts, which can be enormous [1,2] and even irreversible, such as contributing to the extinction of native species [3,4]. This magnitude of impacts is positively correlated with the population size of invasive species [1,5], so monitoring and studying their populations from the early stages of invasion processes are key to the understanding and management of biological invasions [6,7,8]. Moreover, when scientific resources are limited, citizen science can greatly contribute to the improvement in detection and monitoring methods for invasive species [9,10,11] through datasets at different scales of information. However, population dynamics can be complex when invasive species exhibit lag phases before population explosions [12,13], as well as when interactions between invaders, native biota and the recipient environment regulate their population growth [14,15]. These factors, which may complicate both early detection and the design of effective invasive species monitoring programs, can also invalidate the usefulness of citizen science data for monitoring the population trend of these species.

One of the causes of the introduction of non-native species is the international wildlife trade, in which millions of individuals are moved from their native ranges to new areas around the world [16]. Parrots (Order Psittaciformes) are a prime target for the pet trade, which not only contributes to the decline in their natural populations [17] but also to the introduction of 16% of parrot species outside their native distributions [18]. The origin of these introductions is the accidental or deliberate release of individuals kept in captivity that have successfully established populations in the wild [19], mainly in urban environments [20]. However, their population statuses are poorly known [21,22] and studies focused on their long-term population dynamics are scarce [23,24,25,26,27,28,29,30]. The rose-ringed parakeet (*Psittacula krameri*) and the monk parakeet (*Myiopsitta monachus*) are the two most successful invasive parrots globally [18]. While monk parakeets are native to southern South America, rose-ringed parakeets are native to both southern Asia and sub-Saharan Africa [31], although the sources of their introduced populations in Europe are mainly Uruguay and the Indian subcontinent, respectively [32,33]. Overall, their European populations seem to be growing and spreading [34,35] since their first breeding records in the 1960s and 1970s. Nevertheless, the lack of accurate information on their status and population dynamics may not only hinder the early detection and assessment of their impacts but also their feasible management.

Here, we report long-term population censuses and range distributions of rose-ringed and monk parakeets, established in a metropolitan area of southern Spain (city of Seville and surroundings; Figure 1) over nine consecutive years. Studies conducted in this area have assessed the impacts of rose-ringed parakeets on native fauna through the competition for nesting sites [36], with a population decline in a threatened bat species being a key factor [37], while monk parakeets facilitate nesting sites for some native but also non-native species, such as the rose-ringed parakeet [38]. Moreover, both parakeet species show a high prevalence of a novel avian circovirus [39] and disperse seeds from a large number of non-native and invasive plants [40,41]. Thus, our study aims to show the population growth, range expansion and current population sizes of these two species to assess their current status within the invasion process and provide basic information for their management. For these purposes, we combined information from our annual censuses with data collected by two main citizen science platforms, namely eBird and Observation.

## 2. Materials and Methods

### 2.1. Study Area and Species

The study was conducted in the city of Seville and nearby municipalities, an extensive area of 735 km^2^ along the Guadalquivir, Guadiamar and Guadaíra rivers (Figure 1A), encompassing urbanized areas that are surrounded by intensive agriculture (mainly olives, cereals, oranges, and sunflowers). The urban environment has parks and gardens dominated by ornamental trees, such as plane trees (*Platanus* spp.), palms (*Phoenix* spp. and *Washingtonia* spp.), eucalyptus (*Eucalyptus* spp.) and false acacias (*Gleditsia* spp., *Robinia* spp., and *Sophora* spp.). Thus, the breeding areas of both parakeets are concentrated in these green areas, where plane trees are mainly used as nesting and roosting sites by rose-ringed parakeets [36], while date palms (*Phoenix* spp.) are the main nesting substrate for monk parakeets [38]. Although the first records of both parakeet species in Spain date back to the mid-1970s [42,43], their first records in Seville did not appear until the early 1990s. To our knowledge, a small group of rose-ringed and monk parakeets, deliberately released in María Luisa Park in 1992, was the origin of the current established populations in Seville. Since then, their populations have increased, as shown by some population estimations, with ca. 10, 50, 1000 and 1367 rose-ringed parakeets in 2000, 2002, 2011 (P. Edelaar, com. pers.), and 2015, respectively [43,44,45], and ca. 8 and 96 monk parakeets in 2000 and 2015, respectively [42,44].

### 2.2. Field Surveys and Population Censuses

We conducted censuses of both parakeet species from 2013 to 2021 (Figure 1B,C; [36,37,38]), combining surveys of the breeding populations and roost counts. Communal roost counts are recommended to estimate the size of closed populations of parrot species in which all individuals concentrate after breeding in roosts, and all of them can be located [46]. This is the case with rose-ringed parakeets, so we located and monitored all of their communal roosts. Given that this species uses the same roost sites (trees) throughout the year [34], we actively looked for large flocks of individuals early in the morning or half an hour before the sunset, three months before the counts. Once all roosts were detected, we simultaneously counted all birds present (i.e., on the same afternoon, starting to count when the first rose-ringed parakeets arrived until the last individuals entered [47]) after the breeding season (late July) to assess the total population size. Each communal roost was surveyed by one or two persons, depending on its size and the visibility around the trees used for roosting. The monk parakeet, on the other hand, does not congregate in communal roosts but uses its nest (a structure of sticks containing at least one chamber shared by several individuals; Figure 1C) for roosting [48]. Thus, for this species, every year we recorded all nests occupied in the study area just before breeding (between March and early April, when individuals are not building new chambers but attached to these nests). We identified the tree species used as nesting substrates, visually estimated the height of the nest above the ground, and counted all active chamber entries per nest [49,50]. We considered as an active chamber any chamber in which we recorded the entry or exit of monk parakeets, and thus estimated the number of individuals per communal nest through the occupancy rate of active chambers [42,49,50]. As we could not adequately count all individuals per chamber, we extrapolated the occupancy rate of the well-studied population from Barcelona, Spain [49], by multiplying this value (i.e., 1.52 individuals per active chamber) by the number of active chambers recorded each year in our study population (rounding it to the nearest integer). Using the geographic location of these nests, we also estimated the annual size of the breeding area (i.e., number of 5 × 5 km cells) occupied by this species. Monk parakeets build very conspicuous nests and show a high fidelity to their nesting substrates [38,48], which make it easy to locate them and establish the causes of nest abandonment, such as pruning, storms, or tree death. This information was confirmed by birdwatchers (see Acknowledgments), neighbors, or the Seville City Council’s Parks and Gardens Service.

### 2.3. Processing and Validation of Data from Citizen Science

We used citizen science data to compare it with our censuses and estimate the spatial and temporal distribution of both parakeets in the study area. For this purpose, we requested all available occurrence data of rose-ringed and monk parakeets from the two main citizen science platforms used by the ornithological community of Seville: eBird [51] and Observation [52]. Only validated observations corresponding to the period from 1 January 2013 to 31 December 2021 (data required on 18 January 2022) and located in our study area were considered. Validated observations are those that underwent a review process in which a team of volunteer reviewers (experts and regional birdwatchers) verified the quality of the observations submitted to the platform. We also requested all available records of birds in our study area during the same period to calculate the sampling effort (i.e., number of bird observations per year). To avoid the repetition of data, we combined observations from both platforms and grouped checklists into single observations. This was possible because shared lists have the same identifier and, in the case of Observation, observations shared with other platforms, such as eBird, were explicitly shown. For records that only show the presence of a parakeet species but not the number of individuals (6% of all observations), we assigned a single individual. Additionally, we counted the number of 5 × 5 km cells in which parakeets were present over the study period to compare population census data with the spatial distribution of observations recorded in citizen science platforms. For the monk parakeet, we also compared the actual range spread of the species with the spatial distribution of observations recorded in citizen science platforms.

### 2.4. Statistical Analyses

We used generalized linear models (GLM) to evaluate the population trend of both parakeets over time (dependent variable: number of each parakeet species; independent variable: years; negative binomial distribution, log link function). We included year in its linear and quadratic form to differentiate exponential from logistic population growth, respectively. We checked for temporal autocorrelation in census data using the Durbin–Watson test (package *DHARMa* [53]). Then, we estimated the annual population growth rate (*r*) of each species. In the case of exponential growth, we used the standard equation *N_t_* = *N_0_ e^rt^*, solving it as *r* = (ln *N_t_* − ln *N_0_*)/*t*, where *N_0_* is the initial population size, and *N_t_* is the population size at time (*t*). When the population fitted a quadratic temporal trend, we used the modified equation *N_t_* = *N_0_* + *r N_0_* (*K* − *N_0_*/*K*), solved as *r* = ((*N_t_* − *N_0_*)/*N_0_*)) * (*K*/*K* − *N_0_*), where *K* is the carrying capacity (i.e., maximum number of individuals a site can hold). *K* was estimated as the number of individuals at which the population stabilized.

We assessed the temporal trend in the annual mean number of active chambers per monk parakeet nest using the generalized additive model (GAM; package *mgcv* [54]). We used semi-parametric smooth functions (*s*) to fit the non-linear relationship between our dependent variable and years (independent variable). We also used GLM (negative binomial error distribution, log link function) to relate the number of observations and the number of parakeets of each species recorded in citizen science platforms (dependent variables) to our population estimates (independent variables; included in their linear and quadratic forms). The annual number of observations of birds recorded in these platforms was included as an offset (log-transformed) in models to control for the potential effect of changes in sampling effort over years.

To assess the range spread of both parakeets during the study period, we used GLM (Conway–Maxwell–Poisson distribution, log link function, package *glmmTMB* [55]) to relate the number of cells with observations of parakeets recorded in citizen science platforms (dependent variable) to our population estimates (independent variable; included in their linear and quadratic forms). Likewise, we assessed the annual number of cells occupied by monk parakeet nests (independent variable; included in their linear and quadratic forms) to the number of cells with observations of monk parakeets from these platforms. The number of cells with bird observations recorded in citizen science platforms was fitted as an offset variable (log-transformed) to control for changes in sampling effort over years. All statistical analyses were conducted in R v. 4.0.3 [56].

## 3. Results

### 3.1. Censuses and Population Growth Rates

Between 2013 and 2021, we identified ten different communal roosts of rose-ringed parakeets throughout the study period, nine located in the city of Seville and another one in the nearby municipality of Dos Hermanas. The number of roosts simultaneously occupied in a given year ranged from two to five. The longest distance between active roosts was 13 km, while the shortest was 300 m. Roosts were found in trees, mainly in *Platanus × hispanica* (eight roosts), but also in *Brachychiton populneus* (one roost) and *Eucalyptus globulus* (one roost). Our data show that the rose-ringed parakeet population increased from 1200 to 6300 individuals between 2013 and 2021 (Figure 2A, Appendix A), fitting an exponential growth rate (Figure 2A, Appendix A). The average annual population growth rate (*r*) was equal to 0.23 (range: 0.07–0.38) between 2013 and 2021.

During the same period, we recorded a total of 714 different monk parakeet nests just before the breeding season (Figure 3A). The number of nests increased from 33 in 2013 to 474 in 2021 (Figure 3A). These nests were located at an average height above ground of 10.46 (SD = 3.88) m. All nesting substrates were trees of nine different species, two of which (the palms *Phoenix dactylifera* and *P. canariensis*) accounted for 88.7% of the total (Appendix A). The nests remained on the same substrate between years (mean = 3.96 years, range: 1–9 years), and although 338 nests were destroyed during the study period, 38.3% were subsequently rebuilt on the same substrates. However, 33.61% of the recorded nests were lost by 2021, mainly due to removal actions (65.83%), the death of the arboreal substrate by parasites (23.75% of palms were killed by the red palm weevil *Rhynchophorus ferrugineus*, order Coleoptera), or storm toppling (4.58%). Only 5.83% of the nests were abandoned for unknown reasons.

The total number of active chambers in monk parakeet nests increased from 46 to 978 between 2013 and 2021, with a mean across years of 2.11 active chambers per nest (SD = 1.54; range: 1–16) (Figure 3A and Figure 4). Although the mean number of active chambers per nest increased at the beginning of the study period, from 2016 onward, it remained constant and even declined during 2021 (GAM testing a curvilinear time effect: *k* = 3, F = 25.36, *p* = 0.0014, R^2^ = 0.85; Figure 3B, Appendix A). From the number of active chambers, we estimated that the monk parakeet population increased from 70 (±13) to 1487 (±273) individuals during the study period (Figure 2B, Appendix A). While the population increased until 2018, this rise has slowed in recent years, fitting a quadratic temporal trend (logistic growth; Figure 2B, Appendix A). The average annual population growth rate (*r*) was estimated at 0.81 (range: 0.39–1.07) for the whole study period.

From 2013 to 2015, the distribution of all monk parakeet nests was restricted to the city of Seville (Figure 4). In 2017, we detected the first nests outside the city and, since then, distributions of nests have been recorded throughout 13 nearby municipalities. The number of 5 × 5 km cells occupied by monk parakeet nests increased from three to 16 between 2013 and 2021 (Appendix A).

### 3.2. Citizen Science Observations

After filtering, citizen science datasets were composed of 6528 observations (49.9% from eBird and 50.1% from Observation) of rose-ringed parakeets (65.27%, *n* = 4261) and monk parakeets (34.73%, *n* = 2267) (Figure 5A), summing 45,105 individuals (33,149 rose-ringed parakeets and 11,956 monk parakeets; Figure 5B). During this period, 123,702 bird observations (excluding those of rose-ringed and monk parakeets) were uploaded in these platforms. Overall, the number of observations as well as the number of individuals of both parakeets increased from 2013 to 2020, with a marked decline in 2021 (Figure 5). After controlling for the increase in the number of bird observations (i.e., sampling effort), our population estimates obtained through systematic censuses predicted the number of parakeet observation but not the number of individuals recorded in citizen science platforms (Table 1 and Appendix A). It is worth mentioning that the number of observations of parakeets increases with the increasing population size (census data), up to a maximum beyond which subsequent population increases do not lead to a greater number of observations (Appendix A).

Between 2013 and 2014, all citizen science observations of both parakeet species were limited to the city of Seville (rose-ringed parakeets, five cells; monk parakeets, two cells). From 2015 onward, the number of cells in which rose-ringed and monk parakeets were recorded increased to 21 and 19 in 2021, respectively (Figure 6 and Appendix A). Between 2013 and 2021, a total of 31 different cells were occupied by both species (rose-ringed parakeets, 30 cells; monk parakeets, 23 cells). During this period, the mean annual number of observations of parakeets per cell was 22 (SD = 16; rose-ringed parakeets: 25, SD = 17; monk parakeets: 19, SD = 14), while the mean annual number of recorded parakeets per cell was 159 (SD = 109; rose-ringed parakeets: 214, SD = 111; monk parakeets: 101, SE = 75). The number of cells with observations of both parakeets recorded in citizen science platforms increased as the number of individuals censused in the study area increased (Table 1 and Appendix A). Similarly, the number of cells with monk parakeet observations increased, as did the number cells with monk parakeet nests (Table 1 and Appendix A), even after controlling for the increment in the number of cells with bird observations.

## 4. Discussion

Knowing the population status and dynamics of invasive species is critical to understanding their impacts [1,5,57] and delineating management strategies [6,58]. However, accessing long-term population information can be problematic due to the difficulty of detecting some invasive species [13], mainly during their typical lag phases (e.g., parrots; [12,59,60,61]), and the limited resources available for monitoring programs [62]. In this study, we show the rapid growth of the two most successful invasive parrots, the rose-ringed and the monk parakeet, in a large anthropized area. Our data allow us to demonstrate that, after their first likely introduction events in the city of Seville in 1992, the populations of both parakeets rapidly increased, perhaps due to a combination of high survival and fecundity (e.g., breeding attempts, clutch, and fledging successes; [63,64]), such as in other invasive populations. Censuses show that the rose-ringed parakeet is growing exponentially, reaching more than 6000 individuals localized in two main breeding nuclei [36,37]. Monk parakeets, on the other hand, have grown exponentially but seem to have reached a stability of about 1500 individuals, distributed in several breeding nuclei scattered throughout the study area. Population censuses, through counts of individuals at roosts, are suitable for communal roosting species, such as some parrot species [44,65,66,67,68]. However, single counts at communal roosts, such as those performed here, do not allow for the identification of uncertainty in population size estimates related to detection errors, which may lead to the underestimation of real population numbers [67,68]. Despite these limitations, roost counts allow a reasonable lower bound for estimating the total population size (e.g., [66]) and its temporal changes (e.g., [68]) when individuals concentrate in well-known localities, thus reducing the probability of overlooking large flocks during census [68]. In the case of rose-ringed parakeets in Seville, we cannot discard small counting errors that may slightly affect annual population sizes, but they are overweighed by the easy locating of the largest roosts [47]. Likewise, our monk parakeet population estimates may show biased results because we used an estimated number of individuals per chamber (i.e., occupancy rate) from another Spanish population [49]. Although the best approach would have been to use an occupancy rate obtained for our population, the use of alternative occupancy rates from similar environments was feasible when censusing other monk parakeet populations [35]. Whereas counting all individuals that occupy each nest would provide the most reliable estimations of the population size, the exhaustive effort made in detecting all nests during our field surveys, as well as the increasing trend seen over years, likely reduced the error margin of our estimations. Indeed, our intensive fieldwork effort allowed us to cover a larger area and detect more monk parakeet nests than the census conducted in 2015 by Molina et al. [42].

Although our approach is valid for understanding the temporal evolution of each species, direct comparisons between the population sizes of both species in the study area should be carefully carried out due to methodological differences. In the case of rose-ringed parakeets, we conducted direct counts of individuals, while we extrapolated population sizes of monk parakeets through the estimated occupancy rate of nest chambers. Moreover, both censuses were conducted in different periods of their breeding cycles. Whereas the rose-ringed parakeet population was censused when it reaches its annual population peak (i.e., when fledglings joined the rest of population after the breeding season), monk parakeet numbers were estimated before the breeding season when the population was at its annual minimum. Bearing in mind these differences, almost three-quarters of the individuals in the study area in 2021 were rose-ringed parakeets, so they appear to have had different invasion histories. The main factors explaining the successful establishment of non-native parrots are propagule pressure (i.e., the number of introduced individuals; [20,69]) and the origin of introduced individuals, with wild-caught birds being more likely to survive and establish non-native populations than captive-bred ones [20,70]. As both parakeet species introduced in Seville share a wild-caught origin [19], it is likely that there was an asymmetric propagule pressure favorable for rose-ringed parakeets during the early stages of their invasion processes. According to the Convention on International Trade in Endangered Species of Wild Fauna and Flora ([71]), 60,868 rose-ringed and 193,600 monk parakeets of wild origin were imported into Spain from 1975 to 2021 (mostly from 1975 to 2005, after which the EU ban on wild-caught bird prohibited their importation [61]). Although propagule pressure does not seem, therefore, the clearest explanation for the different population sizes observed, the subsequent redistribution of traded individuals between cities could be a potential underlying cause of the population differences observed in different areas. For instance, two of the largest populations of monk parakeets in Spain are established in the cities of Barcelona and Málaga (ca., 3670–5000 individuals; [72,73]), while populations of rose-ringed parakeets there are much smaller (ca., 72–200 individuals; [73,74]). These differences between population sizes can also be a consequence of differences in introduction timing, monk parakeets being introduced later than rose-ringed parakeets in areas where the former species is less abundant than the other [42,43]. Complementarily, differences in population numbers may arise due to disparities in their lag phases (i.e., before reaching exponential population growth [61]). Other unexplored factors, such as predation rate [75,76,77] or nest removal [78] could have also have a negative effect on the population growth rate of monk parakeets during the early stages of the invasion process in Seville. However, given that the monk parakeet currently shows a higher rate of population growth than the rose-ringed parakeet, the former may equal, or even surpass, the population size of the latter in a near future. Moreover, the fact that the cavity-nesting rose-ringed parakeet is limited by the availability of nesting holes [30], while the monk parakeet builds its own nests [38], may facilitate a greater population increase in the latter. One interesting result of our study is that the estimated numbers of monk parakeets were fitted to a logistic growth, in which the population seems to be decelerating its growth in recent years, despite the fact that the number of nests and chambers has increased annually. It is worth mentioning that this population trend is unexpected when the study population is compared with other monk parakeet populations established in Spain, which show exponential growths [35,42,72,73] and did not reach their carrying capacity. Therefore, the current monk parakeet population in Seville could be transitorily in a logistic growth phase that may became exponential. Future monitoring programs, including our own occupancy rate calculated for the study population, are needed to detect potential changes in this population trend. Likewise, it is necessary to continue monitoring, including studies based on marked and tagged individuals, to assess population sizes and identify the driving factors of population dynamics of both parakeet species (e.g., nest site availability, breeding success, and predation pressure).

Citizen science data have largely been used in ecological studies but rarely tested using population census information [79,80,81,82]. Here, we tested the reliability of citizen science data collected in two commonly used platforms, comparing them with data obtained from intensive monitoring programs. The number of rose-ringed and monk parakeet observations increased as the parakeet populations increased, until reaching a plateau after which these citizen science data do not reflect actual population sizes. Moreover, the number of individuals obtained from citizen science platforms did not relate to population numbers. However, the number of cells in which parakeets were recorded followed the increment in the population sizes of both species, and in the case of monk parakeet, also depict its spatial spread. These last results should be interpreted with caution, as in the case of monk parakeets, the spatial distribution of citizen science data depicts the expansion of the breeding population across the study area, while in the case of rose-ringed parakeets, it mainly informs on the larger number of individuals present in the city that may be moving to forage outside. Citizen science data have several biases that may explain these results. Although occurrence records from volunteers can be temporally and spatially biased when the detectability of the species is difficult or survey efforts are inadequate [66,83,84,85], these biases are unlikely in this case, given that parakeets are loud and highly conspicuous [59,86], thus facilitating their detection by birdwatchers. However, volunteers can accidentally introduce sampling bias such as a misidentification between both parakeet species (which is rather unlikely due to their different morphologies and behaviors) or under-reporting, especially when these species become abundant, or simply because they are not interesting to observers [87,88]. Although some observations could still not be uploaded in the platforms given our request was made in early 2022 (18 January), this last issue can mostly explain the decrease in the number of observations and parakeets recorded by citizen science platforms in the last year and the lack of relationship between the number of parakeet observations and their population sizes when the species became abundant. Since most nesting and roosting sites of both parakeets are located in the city of Seville, the probability of reporting them in the less crowded cells is smaller, which can also bias the pattern observed. However, after controlling for the spatial variability in sampling effort, we found a good relationship between the spatial expansion of monk parakeets and the spread predicted by citizen science data. All of these issues must be considered when interpreting citizen science data to minimize errors. Moreover, guidelines about the importance of these observations and strategies to collect data should be incorporated into these platforms to obtain higher quality data [89] and allow them to become important research tools for biological invasion science [8,9,57,90].

Population sizes of invasive species are expected to fluctuate over time [91,92], so long-term monitoring programs are essential to identify changes in population trends (e.g., nonlinear dynamics) or to estimate density-dependent impact rates and design feasible management actions [58,91]. During our study period, populations of rose-ringed and monk parakeets increased by a factor of 5 and 20, respectively, which would exacerbate their impacts. In the case of rose-ringed parakeets, competition for nesting sites through aggressive interactions is responsible for the decline in the largest colonies of a threatened bat species (*Nyctalus lasiopterus*) and a threatened falcon species (*Falco naumanni*; [36,37]), an ecological impact that will become more pronounced as the population of the invasive species continues to increase. For the monk parakeets, its population increase and expansion are expected to provide even more nesting sites for secondary cavity nesters [38], something that could benefit some native species but also promote the transmission of parasites and diseases to them [39,93,94], as well as the further expansion of the rose-ringed parakeet [30]. Moreover, the range expansion of both parakeets outside the urban environment will favor the emergence of little known or unnoticed impacts such as the seed dispersal of non-native and invasive plants [40,41] and the already known agricultural damage (e.g., on fruit trees and sunflower crops; [95,96] authors, unpublished data). Considering that the increment in the population size of both species would intensify their impacts, a management program was implemented in 2019 by the city government of Seville to control the rose-ringed parakeet population, which has to date mainly focused on reducing their breeding success. Moreover, the periodic removal of monk parakeet nests was conducted in several green areas of the city and other municipalities over the study period. Based on our data, these actions were not effective given our results, as rose-ringed and monk parakeet populations have not shown any population reductions but continue to growth. Therefore, alternative methods [97,98,99,100,101,102] should be considered if the responsible authorities aim to effectively manage these populations.

## 5. Conclusions

Monitoring programs focused on populations of invasive species are essential for the understanding of their population dynamics and management. Therefore, we urge the continuation of this monitoring program to provide further information about population trends and the expansion ranges of both parakeet species. Although citizen science data cannot replace monitoring programs, we believe that this activity should be encouraged to improves the environmental education and awareness focused on threats to biodiversity that involve invasive species.

## Figures and Tables

**Figure 1 animals-12-00677-f001:**
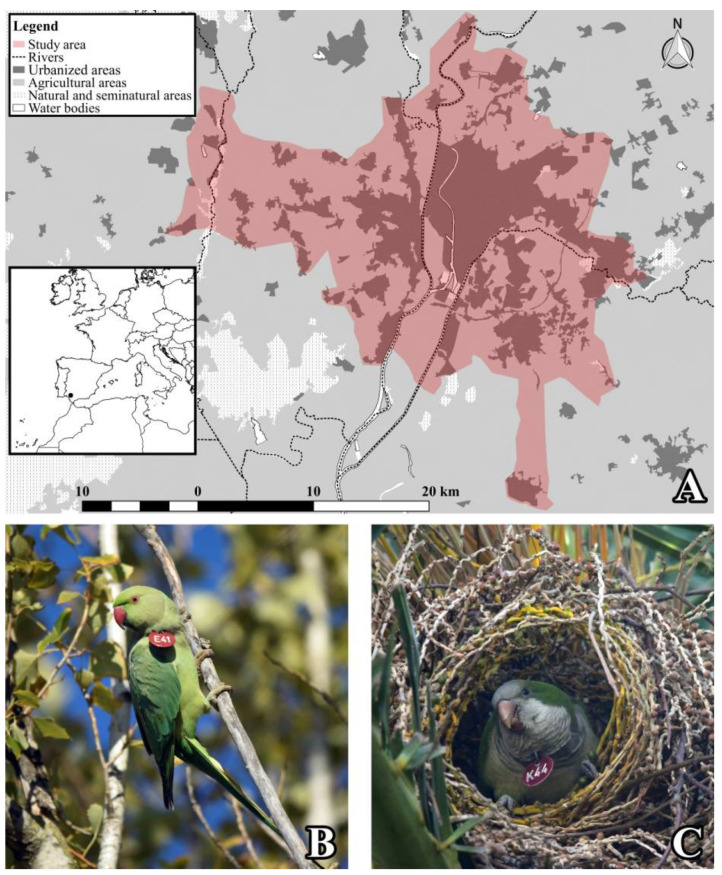
*(***A**) Location of the study area in the metropolitan area of Seville (southern Spain). *(***B**) A rose-ringed (*Psittacula krameri*) and *(***C**) a monk parakeet (*Myiopsitta monachus*) individually marked during our study period. Photos by D. Hernández-Brito.

**Figure 2 animals-12-00677-f002:**
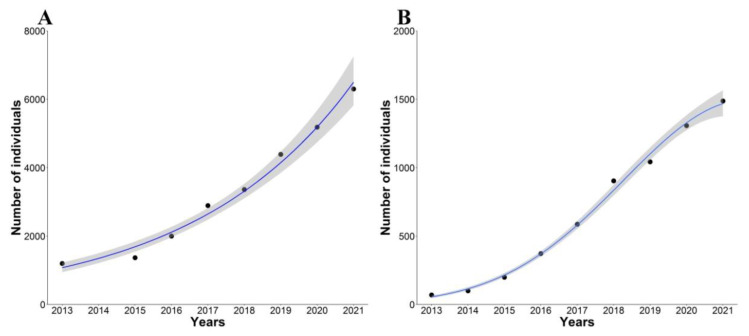
Population trends of (**A**) rose-ringed and (**B**) monk parakeets established in the metropolitan area of Seville from 2013 to 2021. Blue lines show model fit estimated throughout GLM (negative binomial error distribution, log link function). Grey area: 95% confidence interval.

**Figure 3 animals-12-00677-f003:**
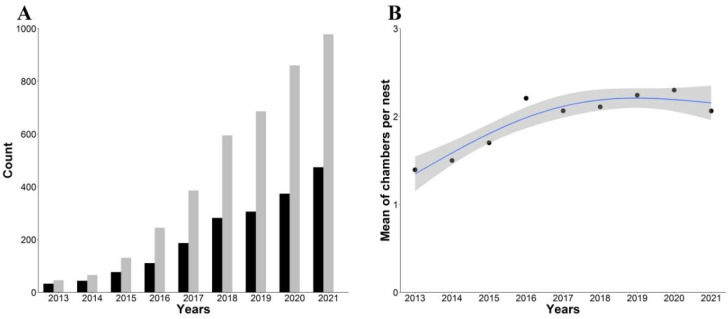
(**A**) Number of nests (black bars) and total number of active chambers (grey bars) of monk parakeets censused in the metropolitan area of Seville from 2013 to 2021, and (**B**) mean number of active chambers per nest. The blue line shows the fitted generalized additive model (grey area: 95% confidence interval).

**Figure 4 animals-12-00677-f004:**
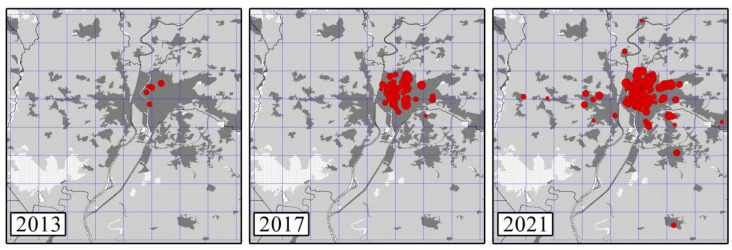
Distribution of monk parakeet nests (red dots) in the study area over years. The size of dots increases with the number of chambers per nest (from 1 to 16).

**Figure 5 animals-12-00677-f005:**
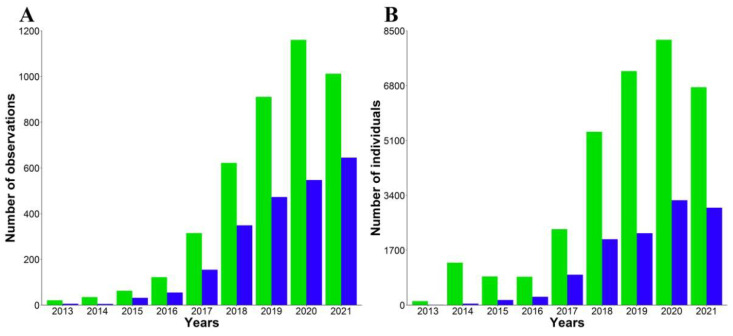
Number of (**A**) observations and (**B**) individuals of rose-ringed (green bars) and monk parakeets (blue bars) annually recorded on two citizen science platforms (eBird and Observation) in the study area from 2013 to 2021.

**Figure 6 animals-12-00677-f006:**
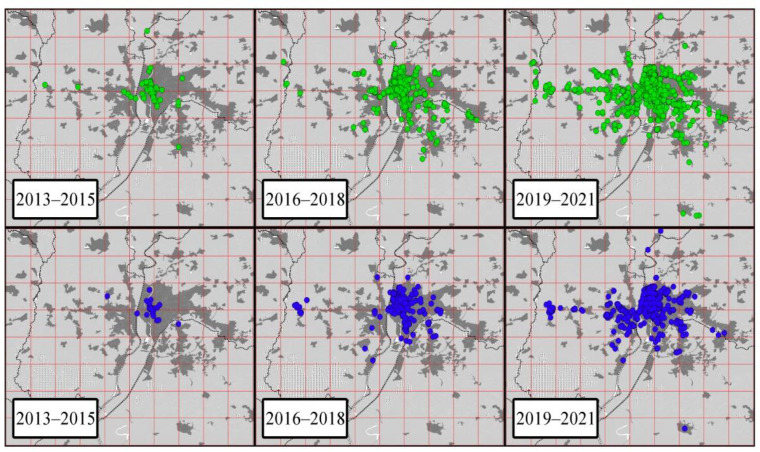
Observations of rose-ringed (green dots) and monk parakeets (blue dots) recorded in two citizen science platforms (eBird and Observation) in the study area over three consecutive year intervals between 2013 and 2021.

**Table 1 animals-12-00677-t001:** Models obtained to relate the number of rose-ringed and monk parakeet observations and individuals recorded in the citizen science platforms to our annual population counts (Survey) conducted in the metropolitan area of Seville from 2013 to 2021. The number of bird observations recorded in these platforms was fitted as an offset to control for changes in sampling effort over years. We also include models obtained to relate the number of cells with rose-ringed and monk parakeet observations recorded in these platforms to the annual population counts and the number of cells with monk parakeet nests (Nests). The number of cells with bird observations recorded in these platforms was fitted as an offset to control for changes in sampling effort over years. Dev. Expl.: deviance explained. *p*-values in bold font indicate statistical significance (≤0.05). See Appendix A for outputs of all the models that were run.

**Rose-Ringed Parakeets**						
**Dependent Variables**	**Independent Variable**	**Estimate**	**SE**	** *z* **	** *p* **	**Dev. Expl.**
Number of observations	Intercept	−3.43	0.04	−82.4	**<0.0001**	46.33%
	Survey	0.16	0.12	1.29	0.19	
	Survey^2^	−0.3	0.11	−2.8	**0.005**	
Number of individuals	Intercept	−1.19	2.8	−4.23	**<0.0001**	8.51%
	Survey	<−0.0001	<0.0001	−0.62	0.54	
Number of cells	Intercept	−1	0.06	−16.97	**<0.0001**	62.26%
	Survey	0.98	0.17	5.28	**<0.0001**	
	Survey^2^	−0.41	0.15	−2.69	**0.008**	
**Monk Parakeets**						
**Dependent Variables**	**Independent Variable**	**Estimate**	**SE**	** *z* **	** *p* **	**Dev. Expl.**
Number of observations	Intercept	−4.2	0.06	−6.6	**<0.0001**	52.32%
	Survey	0.63	0.2	*3.2*	**<0.002**	
	Survey^2^	−0.35	0.15	−2.41	**0.02**	
Number of individuals	Intercept	−2.85	0.26	−18.98	**<0.0001**	16.22%
	Survey	0.0005	0.0003	1.48	0.137	
Number of cells	Intercept	−2.12	0.14	−14.54	**<0.0001**	58.12%
	Survey	0.0009	0.0001	6.63	**<0.0001**	
Number of cells	Intercept	−2.24	0.23	−9.78	**<0.0001**	68.1%
	Nests	0.09	0.02	4.57	**<0.0001**	

## Data Availability

The raw data supporting the findings of this article have been uploaded as Appendix A. eBird data can be downloaded from here: [51]. Observation data can be requested from here: [52]. The rest of the raw data are provided in the body of the article.

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
