# Peer review of "Annual Censuses and Citizen Science Data Show Rapid Population Increases and Range Expansion of Invasive Rose-Ringed and Monk Parakeets in Seville, Spain"

_animals, 2022, doi:10.3390/ani12060677_

Round 1
Reviewer 1 Report
This was a very enjoyable manuscript to review. It’s well-written and the science appears sound. The authors have published several papers on this system and do a nice job of weaving in their previous research.
All of my comments are minor, focusing on a couple suggestions for the figures, minor phrasing issues, and suggesting three citations that may be worth incorporating into the Discussion.
Figures
Stacked bar charts are usually used to break down and compare parts of a whole. In contrast, Figure 2B shows three separate categories and requires the reader to do a bit of math in order to determine the number of active chambers per monk parakeet nest (the gray bars) and the number of individuals (black bars). I think that it makes more sense to have a standalone figure with three parts (number of individuals represented in section a, number of nests in section b, number of active chambers per nest in section c).
The same issue applies to 4D; it’s a bit complicated to understand, particularly given that there are two y-axes. It would be easier to read if 4D was split into multiple sections.
Minor phrasing issues
On line 196, the spacing before the word “Between” is odd. Perhaps this was meant to be the start of a standalone paragraph?
Line 215-216; generally the format is day-month-year
Lines 306-309. I couldn’t quite make sense of what the end of this sentence was saying. Could it be reworded for clarity?
Line 323. I’m not sure that “critically reduces” makes sense here. Perhaps “sharply reduces”?
Additional citations
The authors wrote an additional chapter in the “Naturalized Parrots of the World” book that would be relevant to include in this manuscript. It seems like it would fit well into the Discussion. The citation is
• Carrete et al. 2021; Carrete, M.; Abellán, P.; Anadón, J.D.; Tella, J.L. The fate of multistage parrot invasions in Spain and Portugal. In Naturalized Parrots of the World: Distribution, Ecology, and Impacts of the World's Most Colorful Colonizers; Pruett-Jones, S., Eds.; Princeton University Press: Princeton, NJ, USA, 2021; pp. 240–248.
It might also be worth mentioning a couple other papers from elsewhere in Europe that seem like they would fit with the topics raised in the Discussion.
• Butler et al. 2013. Butler, C. J.; Cresswell, W.; Gosler, A.; Perrins, C. The breeding biology of Rose-ringed Parakeets Psittacula krameri in England during a period of rapid population expansion. Bird Study 60(4): 527-532. This paper mentions the relatively high fecundity of rose-ringed parakeets in an introduced population, which fits in well with the discussion about the rapid increase for this species noted in this manuscript.
• Shwartz et al. 2009. Shwartz, A.; Strubbe, D.; Butler, C. J.; Matthysen, E.; Kark, S. The effect of enemy-release and climate conditions on invasive birds: a regional test using the rose-ringed parakeet (Psittacula krameri) as a case study. Diversity and Distributions 15: 310-318. This might be worth mentioning alongside the propagule hypothesis in the Discussion.
Author Response
Reviewer 1:
This was a very enjoyable manuscript to review. It’s well-written and the science appears sound. The authors have published several papers on this system and do a nice job of weaving in their previous research.
All of my comments are minor, focusing on a couple suggestions for the figures, minor phrasing issues, and suggesting three citations that may be worth incorporating into the Discussion.
Thank you very much for these positive comments, we appreciate very much all the detailed suggestions provided. We have addressed all of them since we feel they are helping us to improve the clarity of our manuscript. Changes can be easily seen in the new version with tracked changes.
Figures
Stacked bar charts are usually used to break down and compare parts of a whole. In contrast, Figure 2B shows three separate categories and requires the reader to do a bit of math in order to determine the number of active chambers per monk parakeet nest (the gray bars) and the number of individuals (black bars). I think that it makes more sense to have a standalone figure with three parts (number of individuals represented in section a, number of nests in section b, number of active chambers per nest in section c).
As both reviewers agree on this point, we have divided the Figure 2 into two different figures:
Figure 2. Population trends of A) rose-ringed and B) monk parakeets established in the metropolitan area of Seville from 2013 to 2021. Plots represent censused (black dots) and estimated population sizes (i.e., population sizes expected considering the estimated annual population growth rate r (green line: minimum r, blue line: mean r, red line: maximum r).
Figure 3: Number of A) nests and B) total number of active chambers of monk parakeets in the metropolitan area of Seville from 2013 to 2021.
The same issue applies to 4D; it’s a bit complicated to understand, particularly given that there are two y-axes. It would be easier to read if 4D was split into multiple sections.
Following the reviewer suggestion, we have divided Figure 4 into two different figures (which have changed their numbering after splitting Figure 2):
Figure 5. Observations of rose-ringed (green dots) and monk parakeets (blue dots) recorded in two citizen science platforms (eBird and Observation) in the study area over three consecutive year intervals between 2013 and 2021.
Figure 6. Number of A) observations and B) individuals of rose-ringed (green bars) and monk parakeets (blue bars) annually recorded on two citizen science platforms (eBird and Observation) in the study area from 2013 to 2021.
Minor phrasing issues
On line 196, the spacing before the word “Between” is odd. Perhaps this was meant to be the start of a standalone paragraph?
Thanks for noting us this mistake. We have correct it.
Line 215-216; generally the format is day-month-year
Done.
Lines 306-309. I couldn’t quite make sense of what the end of this sentence was saying. Could it be reworded for clarity?
We have reformulated the sentence to clarify it (lines 417-420).
Line 323. I’m not sure that “critically reduces” makes sense here. Perhaps “sharply reduces”?
We have deleted this part of the discussion following the second referee’s suggestion.
Additional citations
The authors wrote an additional chapter in the “Naturalized Parrots of the World” book that would be relevant to include in this manuscript. It seems like it would fit well into the Discussion. The citation is
• Carrete et al. 2021; Carrete, M.; Abellán, P.; Anadón, J.D.; Tella, J.L. The fate of multistage parrot invasions in Spain and Portugal. In Naturalized Parrots of the World: Distribution, Ecology, and Impacts of the World's Most Colorful Colonizers; Pruett-Jones, S., Eds.; Princeton University Press: Princeton, NJ, USA, 2021; pp. 240–248.
It might also be worth mentioning a couple other papers from elsewhere in Europe that seem like they would fit with the topics raised in the Discussion.
- Butler et al. 2013. Butler, C. J.; Cresswell, W.; Gosler, A.; Perrins, C. The breeding biology of Rose-ringed Parakeets Psittacula krameri in England during a period of rapid population expansion. Bird Study 60(4): 527-532. This paper mentions the relatively high fecundity of rose-ringed parakeets in an introduced population, which fits in well with the discussion about the rapid increase for this species noted in this manuscript.
- Shwartz et al. 2009. Shwartz, A.; Strubbe, D.; Butler, C. J.; Matthysen, E.; Kark, S. The effect of enemy-release and climate conditions on invasive birds: a regional test using the rose-ringed parakeet (Psittacula krameri) as a case study. Diversity and Distributions 15: 310-318. This might be worth mentioning alongside the propagule hypothesis in the Discussion.
Thank you for your suggestions, we have added these references in the Discussion (lines 317-329; 350-357).
Reviewer 2 Report
The authors show the results of nine years of monitoring of rose-ringed and monk parakeet populations in Seville (Spain). They also gathered data from citizen science projects and compared this information with their own records. The authors found a rapid and exponential growth of populations of both invasive species. Along with this increase, they also found a spread of both species in the surrounding urban areas of Seville.
Overall, the paper is well presented and written. The objectives and results are timely and necessary for an evidence-based management of these pests. Overall, the main findings agree with what anyone would expect for these invasive birds, based on their trajectory in other European cities. However, I have a number of major concerns about some results. Perhaps, a more detailed description of the methods will suffice to solve some of these problems (see my minor comments below).
Main comments:
- There was an important difference between rose-ringed and monk parakeet population surveys: rose-ringed parakeets were counted just after the breeding season, when population reaches its annual peak (adults + juveniles); while monk parakeets nests were surveyed just before the breeding season, when population has its annual minimum (only adults who surveyed from the previous breeding season + recruits). (Note: I am assuming Seville populations are closed and do not receive individuals from far populations). Of course, this difference does not matter to estimate temporal trends for each population. However, it may be important when comparing absolute numbers of both species. Moreover, rose-ringed parakeet data is a true census, as you counted all individuals, while the number of monk parakeet data is an estimation from nests. The (unknown) uncertainty related to the latter number is poorly recognized in the paper and both population values are considered equally real.
- Spatial analysis. The 5x5 km cell approach is ok, but I think that the authors should do a more refined approach by using basic spatial analyses, such as minimum convex polygons or kernel densities. As the authors have the exact locations of the observations, they can estimate accurate distribution areas. As urban areas (ie suitable habitat) are not available homogeneously across the study area, the cell approach is quite poor and uninformative. Moreover, as the sampling effort was not equal in all cells, presence/absence in them cannot be compared.
- Results from citizen science data. They provide a similar pattern to the one provided by your accurate censuses. However, in my opinion, this may be just a nice coincidence. The exponential growth in the number of observations/individuals provided by citizen science data can be showing simply the exponential growth of the observers’ activity in both projects. There are two results supporting my interpretation: 1) Based on your surveys, monk parakeets increased ~5 times more than rose-ringed parakeets (425% vs 2024%). However, based on citizen science, both species increased a similar amount (6787 vs 6995%). Perhaps, this ~7000% is just the % increase of use of both citizen science platforms from 2013 to 2020. 2) The magnitude of the population increase suggested by the citizen science projects would be extremely overestimated. For instance, compare 425 vs 6787 (16-fold!). In sum, if you do not account for the sampling effort, the total number of records is a meaningless measure. As the sampling effort can be hardly known in non-standardized monitoring schemes, as ebird or observation, you may use the total number of records per year as a proxy for the effort. Alternatively, the % of complete lists reporting the species may be a preferable approach. However, more complex approaches based on state-space models to account for imperfect detection would be the right choice for your analysis.
Minor comments:
- L2: Title. I suggest to shorten it. For instance, I would remove: “Annual censuses and citizen science data show”. In general, the authors put too much emphasis on the methods and data sources. I am aware that “citizen science” is currently a very fashionable topic, but I would not do an excessive marketing of these key words. Focus on the relevant ecological objectives and findings of your study: invasive parakeets and their population growth.
- L26: Sure? I think these species have been subjected to monitoring in other cities, such as Barcelona or Malaga, before than 2013.
- L74: Remove “first”.
- L106: Fig.1A. What is an “artificial area”? Perhaps urban areas would be more suitable.
- L123: Here, you may cite your fig. 1c.
- L131: This value is very important, but the authors did not validate it in their own population after 9 years of study. This is quite surprising…
- L133-35: The results from this analysis are missing.
- L143: What was a “validated observation”? How were observations validated? Who did it? I know that both citizen science projects have local birders that review all records. I assume you are referring to this validation process. If this is the case, please explain it, including details on the procedures used by these birders during validation.
- L145: How did you combine both datasets? Ebird is based on lists, while observation is (mainly) based on single observations. Honestly, I can hardly imagine how you detected duplicated records. This methods section needs more detailed explanations.
- L148: Although I can accept this assumption, I would suggest a double analysis: one with those records with count data, and another with all records just considered as presences.
- L153-158: Another example of poor methodology explanation. Please, explain better how did you calculated r.
- L160-163: Please, exclude 2021 from your analyses. It does not make sense, if this year is incomplete. Otherwise, you can update your database and download data from Nov and Dec.
- L166-167: This sentence is methods.
- L168-169: How did you find all the roosts? I mean, how are you sure that you found all rose-ringed roosts in a so big study area? Perhaps, a more detailed explanation on the field methods would be necessary.
- L174: Are these numbers right? If you counted all the roosting birds, I would expect less rounded values. A supplementary table with the annual values of counted parakeets would be welcome.
- L175: Remove “representing a population increase of 425% in nine years”. It is a non-intuitive way to show the same result again.
- L178: I can hardly trust on a perfect correlation (r=1). Looking at the fig. 1a, I would expect r<1. Moreover, if r=1, then p should be 0.
- L179: Fig. 2b. Each category should be in a separated graph. In any case, black bars (num ind) should be 1.52 times the grey bars (num chambers). However, they are equal. I think there is a mistake.
- L183: Remove “per monk parakeet nest”.
- L185: As previously, a supplementary table with the raw data per year would be welcome.
- L186: Remove “showing an increase of 1336.36% over nine consecutive breeding seasons.”. Same justification than before.
-L187-195: These are very interesting results, but you explained in the methods nothing about other information related to the nest than its location and num of chambers.
- L189: Please, provide a detailed table for all tree species.
- L190-1: If you visited nests once a year, how can you know whether the nests were destroyed, abandoned, etc.? As I suggested above, you need to give more details on your methodology.
-L196: An interesting metric would be the annual average of chambers per nest. This value may give information on the temporal evolution of nests’ structure.
- L200: Remove “showing an increase of 433.33% during this period”
- L202: Remove “(increase of 2026.09%)”
- L205: Remove: “estimated population increase of 2024.28%”. This % should be equal to the previous one as the number of individuals = number of chambers*1.52. Can you justify this discrepancy?
- L207-208: Same comment as above, I can hardly trust on a r=1.
- L209: As you did not explain how you looked for and found the nests in the methods, I could argue that Fig. 3 is due to an increasing sampling effort.
- L213: Did you filter data? Please, explain in the methods.
- L214-215: 4135+2197=6332. This is not 5930, why?
- L219: Respectively to …?
- L221-223: Comparing fig. 4d and fig. 2, a perfect correlation is not possible. Please, check these results.
- L223-25: As suggested above, I would remove any analysis with the incomplete year 2021.
- L234: Remove “a increase of 500% between 2013 and 2021”.
- L247-248: You cannot claim this, as you do not have info on the population between 1992 and 2012.
- L264-265: Please, give updated population numbers (e.g., check the new Catalan atlas). On the other hand, I think that monk parakeets arrived to Madrid and Barcelona many years before than rose-ringed parakeets. Monk parakeets have larger populations there simply because they have had more years to thrive.
- L274: There are alternative hypotheses to understand the current situation of parakeets in Seville. 1) You know nothing on what happened between 1992 and 2012 (20 years!). Perhaps, the monk parakeets released in 1992 went extinct few years later and the current population was reintroduced later. 2) The monk parakeet is subjected to pest control by removing their nests. Perhaps, these actions constrained population growth in early stages. 3) Finally, as I explained above, the numbers provided for both species are not fully comparable, as they come from the maximum and minimum annual population size.
- L293-294: I think the argument should be in the opposite direction. Your data supports citizen science data, not the reverse. However, see my main comments.
- L298: I agree that both species are easily detected in the field, however there are other potential issues overlooked by the authors. 1) Misidentification of both species (plausible for less experienced birders). 2) As they are exotic species, some birders may decide to avoid reporting them. 3) Sampling effort was not homogeneous in your study area. The probability of reporting them in the less crowded cells of your study area is smaller. Thus spatial spread could be representing observers’ presence rather than the true pattern of species expansion.
- L303-346: This is a nice discussion, but I think it is far from the objectives, data and results of this study. However, I did not see any discussion about the spatial spread of the species.
- L348-350: I would remove the first sentence of the conclusions.
Author Response
The authors show the results of nine years of monitoring of rose-ringed and monk parakeet populations in Seville (Spain). They also gathered data from citizen science projects and compared this information with their own records. The authors found a rapid and exponential growth of populations of both invasive species. Along with this increase, they also found a spread of both species in the surrounding urban areas of Seville.
Overall, the paper is well presented and written. The objectives and results are timely and necessary for an evidence-based management of these pests. Overall, the main findings agree with what anyone would expect for these invasive birds, based on their trajectory in other European cities. However, I have a number of major concerns about some results. Perhaps, a more detailed description of the methods will suffice to solve some of these problems (see my minor comments below).
Thank you very much for these positive comments. We appreciate very much all the detailed suggestions provided. We have addressed all of them since we feel they are helping us to improve the clarity of our manuscript. Changes can be easily seen in the new version with tracked changes.
Main comments:
- There was an important difference between rose-ringed and monk parakeet population surveys: rose-ringed parakeets were counted just after the breeding season, when population reaches its annual peak (adults + juveniles); while monk parakeets nests were surveyed just before the breeding season, when population has its annual minimum (only adults who surveyed from the previous breeding season + recruits). (Note: I am assuming Seville populations are closed and do not receive individuals from far populations). Of course, this difference does not matter to estimate temporal trends for each population. However, it may be important when comparing absolute numbers of both species. Moreover, rose-ringed parakeet data is a true census, as you counted all individuals, while the number of monk parakeet data is an estimation from nests. The (unknown) uncertainty related to the latter number is poorly recognized in the paper and both population values are considered equally real.
We perfectly understand this comment and we have extended this point discussing the differences between both approaches to estimate the number of rose-ringed and monk parakeets in our study (lines 327-332).
- Spatial analysis. The 5x5 km cell approach is ok, but I think that the authors should do a more refined approach by using basic spatial analyses, such as minimum convex polygons or kernel densities. As the authors have the exact locations of the observations, they can estimate accurate distribution areas. As urban areas (ie suitable habitat) are not available homogeneously across the study area, the cell approach is quite poor and uninformative. Moreover, as the sampling effort was not equal in all cells, presence/absence in them cannot be compared.
This is the only suggestion where we do not agree with the reviewer. As we recognize in the manuscript, we just attempt to show a first, roughly approach to the range spread of the two parakeet species. In the case of the nests of monk parakeets monitored by our research team (new Figure 4), we think that the use of cells is adequate. We show the exact location of each nest and a measure of density (the number of chambers in each nest with points increasing in size). However, plotting the observations recorded in citizen science platforms (new Figure 5) is more questionable, as these observations are subject to several sources of bias that are difficult to identify and correct (now discussed in lines 391-407). Moreover, we conservatively considered the presence of one individual for those observations of the presence of the species where the number of individuals was not indicated by the observer (lines 173-175), while actually the observation could correspond to large flocks. Moreover, observations do not correspond to nests but to individuals or flocks that can be foraging or simply flying to cover large distances from nesting/roosting sites to foraging areas. Altogether, the use of more refined approaches would suffer from the same sources of bias. Despite of all, we still think our approach allows us to show a first, although imperfect, picture of the range expansion of the species using citizen science, with the caveats recognized in Discussion.
On the other hand, we are almost ready to initiate a two-year telemetry project using a large number of tagged individuals from each parakeet species. The information obtained will allow us knowing the exact location of each individual several times at day. It will offer us an invaluable and unbiased information on the movements, use of habitats and space of the two species that will be adequate for the finer spatial analyses suggested by the reviewer.
- Results from citizen science data. They provide a similar pattern to the one provided by your accurate censuses. However, in my opinion, this may be just a nice coincidence. The exponential growth in the number of observations/individuals provided by citizen science data can be showing simply the exponential growth of the observers’ activity in both projects. There are two results supporting my interpretation: 1) Based on your surveys, monk parakeets increased ~5 times more than rose-ringed parakeets (425% vs 2024%). However, based on citizen science, both species increased a similar amount (6787 vs 6995%). Perhaps, this ~7000% is just the % increase of use of both citizen science platforms from 2013 to 2020. 2) The magnitude of the population increase suggested by the citizen science projects would be extremely overestimated. For instance, compare 425 vs 6787 (16-fold!). In sum, if you do not account for the sampling effort, the total number of records is a meaningless measure. As the sampling effort can be hardly known in non-standardized monitoring schemes, as ebird or observation, you may use the total number of records per year as a proxy for the effort. Alternatively, the % of complete lists reporting the species may be a preferable approach. However, more complex approaches based on state-space models to account for imperfect detection would be the right choice for your analysis.
We thank very much the reviewer for this comments, as we have not clarified in the text that the relationship between the population growth of parakeets and the number of observations in citizen science platforms was not the result of an increment in sampling effort. Thus, in the new version of the ms, we have included linear models instead of correlations where we incorporated a surrogate of sampling effort (i.e., number of observers per year) as a controlling variable. We have detailed this point in Materials and Methods (lines 187-194) as well as in Results (lines 212-214, lines 255-257, new Table 1).
Minor comments:
- L2: Title. I suggest to shorten it. For instance, I would remove: “Annual censuses and citizen science data show”. In general, the authors put too much emphasis on the methods and data sources. I am aware that “citizen science” is currently a very fashionable topic, but I would not do an excessive marketing of these key words. Focus on the relevant ecological objectives and findings of your study: invasive parakeets and their population growth.
We have mixing feelings about this minor comment. We are aware our population monitoring results are the strongest part of the paper, while results from citizen science are complementary and –unavoidably- more methodologically questioned. Therefore, it seems reasonable to remove citizen science from the title. However, on the other hand, citizen science has been increasingly used to model distributions and population sizes for a variety of organisms, while lacking real data from these populations to contrast/validate the information derived from citizen science. In this sense, we show here that trends obtained from citizen science are -at least roughly- supported by accurate monitoring programs. We think this point would attract a wide range of researchers and managers, interested on the use of citizen science, much larger than the number of readers just interested on our study species. In the light of these thoughts, we would prefer to maintain citizen science in the title, but we are open to remove it if the reviewer or editor find it more adequate.
- L26: Sure? I think these species have been subjected to monitoring in other cities, such as Barcelona or Malaga, before than 2013.
The comment of the reviewer is correct and we have corrected this sentence accordingly (by removing "the first").
- L74: Remove “first”.
Done.
- L106: Fig.1A. What is an “artificial area”? Perhaps urban areas would be more suitable.
We used the category "artificial area" to include urban, suburban, and industrial surfaces. As suggested by the reviewer, we have renamed the category as "urbanized areas".
- L123: Here, you may cite your fig. 1c.
Done.
- L131: This value is very important, but the authors did not validate it in their own population after 9 years of study. This is quite surprising…
We understand the comment of the reviewer, and the truth is that we have some counts made in a sample of nests not representative at all of the whole population. However, given that our field surveys required a great effort to cover the study area (735 km2) every year and our monitoring resources are really limited, we were not able to estimate the occupation rate of our whole study population.
- L133-35: The results from this analysis are missing.
These results are shown in lines (246-250) and lines (297-307) in Results, as well as in Figure S1.
- L143: What was a “validated observation”? How were observations validated? Who did it? I know that both citizen science projects have local birders that review all records. I assume you are referring to this validation process. If this is the case, please explain it, including details on the procedures used by these birders during validation.
Effectively, this is the validation process. We have added a detailed explanation about that in lines 165-170.
- L145: How did you combine both datasets? Ebird is based on lists, while observation is (mainly) based on single observations. Honestly, I can hardly imagine how you detected duplicated records. This methods section needs more detailed explanations.
We understand the confusion of reviewer. We have explained in the new version of the manuscript how we combined both datasets. Briefly, when you request a dataset from eBird, all observations of a species are showed as single observations, not included in a list with the rest of observations of other species. Regarding detection of duplicated records, we were referring those observations present in both citizen science projects as well as those observations that are shared by different birdwatchers that are observing birds together. These cases can be detected through ID lists and Observation can filter those observations present in other projects, such as eBird. Following the reviewer recommendation, we have extended explanations on the filtering procedure (lines 170-173).
- L148: Although I can accept this assumption, I would suggest a double analysis: one with those records with count data, and another with all records just considered as presences.
We understand the concern of the reviewer about our asumption. However, as the number of occurrence data is very low (ca. 6.4%), our analysis are not biased when we conservatively considered a single individual for occurrence observations. Indeed, we have performed models using as dependent variable the total number of individuals obtained including (Model 1) or not (Model 2) the 6% of observations without explicit information about abundances and results remained similar for the independent variable (Population count):
Rose-ringed parakeet:
- Model 1: Population count (estimate = 1.09; SE = 0.46; t value = 1.41, p =0.06)
- Model 2: Population count (estimate = 1.1; SE = 0.46; t value = 2.39, p =0.06)
Monk parakeet:
- Model 1: Population count (estimate = 2.61; SE = 0.37; t value = 7.03, p =0.0004)
- Model 2: Population count (estimate = 2.62; SE = 0.37; t value = 7.16, p =0.0004)
Therefore, we use models 2 to test relations between the number of recorded individuals to our population counts (new Table 1). To avoid confusions among readers, we have shown the percentage of the occurrence data in the citizen science dataset in the Results (line 268).
- L153-158: Another example of poor methodology explanation. Please, explain better how did you calculated r.
We have rewritten this part for clarity (lines 180-185).
- L160-163: Please, exclude 2021 from your analyses. It does not make sense, if this year is incomplete. Otherwise, you can update your database and download data from Nov and Dec.
We agree on this point and we have updated our database, so all analyses focused on citizen science data now include 2021 as full year.
- L166-167: This sentence is methods.
We have deleted the sentence to avoid this redundancy.
- L168-169: How did you find all the roosts? I mean, how are you sure that you found all rose-ringed roosts in a so big study area? Perhaps, a more detailed explanation on the field methods would be necessary.
We agree that these aspects of monitoring need clarification in the section of "Field surveys and population censuses". We have redone this part for clarity (lines 124-130).
- L174: Are these numbers right? If you counted all the roosting birds, I would expect less rounded values. A supplementary table with the annual values of counted parakeets would be welcome.
All numbers are correct and we have added a supplementary table (Table S1) that shows the annual values of counted rose-ringed and monk parakeets during our study period.
Table S1. Annual population sizes of rose-ringed and monk parakeets established in the metropolitan area of Seville from 2013 to 2021.
- L175: Remove “representing a population increase of 425% in nine years”. It is a non-intuitive way to show the same result again.
Done.
- L178: I can hardly trust on a perfect correlation (r=1). Looking at the fig. 1a, I would expect r<1. Moreover, if r=1, then p should be 0.
We have finally discarded our analysis of correlation. Alternatively, we have visually assessed the population curves expected based on the annual population growth rate (r). We have clarified that point in the Materials and Methods (lines 180-185) as well as their results (lines 212-214, lines 254-257, Figure 2).
- L179: Fig. 2b. Each category should be in a separated graph. In any case, black bars (num ind) should be 1.52 times the grey bars (num chambers). However, they are equal. I think there is a mistake.
As both reviewers agree on this point, we have divided the Figure 2 into two different figures:
Figure 2. Population trends of A) rose-ringed and B) monk parakeets established in the metropolitan area of Seville from 2013 to 2021. Plots represent censused (black dots) and estimated population sizes (i.e., population sizes expected considering the estimated annual population growth rate r (green line: minimum r, blue line: mean r, and red line: maximum r).
Figure 3: Number of A) nests and B) total number of active chambers of monk parakeets in the metropolitan area of Seville from 2013 to 2021.
Regarding the number of individuals and chambers, they are correct and the new figure (Figure 3A) shows clearly their values.
- L183: Remove “per monk parakeet nest”.
Done.
- L185: As previously, a supplementary table with the raw data per year would be welcome.
Accordingly, we have added a supplementary table (Table S1) that shows the annual values of counted rose-ringed and monk parakeets (see above).
- L186: Remove “showing an increase of 1336.36% over nine consecutive breeding seasons.”. Same justification than before.
Done.
-L187-195: These are very interesting results, but you explained in the methods nothing about other information related to the nest than its location and num of chambers.
We also agree that these aspects of monitoring need clarification in the section of Field surveys and population censuses. We have redone this part for clarity (lines 136-139; 151-155).
- L189: Please, provide a detailed table for all tree species.
We have added a supplementary table (Table S2) that shows the proportion of tree species used as nesting substrate by monk parakeets over our study period.
Table S2: Number of trees of different species used as nesting substrate by monk parakeets in the metropolitan area of Seville between 2013 and 2021.
- L190-1: If you visited nests once a year, how can you know whether the nests were destroyed, abandoned, etc.? As I suggested above, you need to give more details on your methodology.
We also agree that these aspects of monitoring need clarification in the section of Field surveys and population censuses. We have redone this part for clarity (lines 151-155).
-L196: An interesting metric would be the annual average of chambers per nest. This value may give information on the temporal evolution of nests’ structure.
We agree in this point and we have performed analysis to test the temporal trend of mean chambers per nests. We have clarified that point in the Materials and Methods (lines 187-189) as well as their results (lines 250-252, new Figure 3B).
- L200: Remove “showing an increase of 433.33% during this period”
Done.
- L202: Remove “(increase of 2026.09%)”
Done.
- L205: Remove: “estimated population increase of 2024.28%”. This % should be equal to the previous one as the number of individuals = number of chambers*1.52. Can you justify this discrepancy?
We have removed the percentage. Regarding differences between percentages, this discrepancy is due to the fact that values of the number of individuals were rounded to the nearest integer. We have clarified this point in Materials and Methods (lines 148-149).
- L207-208: Same comment as above, I can hardly trust on a r=1.
As detailed above, we have finally discarded our analysis of correlation. Alternatively, we have visually assessed population curves expected per species based on r. We have clarified that point in the Materials and Methods (lines 181-186) as well as in Results (lines 212-214, lines 257-258, Figure 2).
- L209: As you did not explain how you looked for and found the nests in the methods, I could argue that Fig. 3 is due to an increasing sampling effort.
We understand the reviewer's comment, so we have clarified it in the section of Field surveys and population censuses (lines 134-155).
- L213: Did you filter data? Please, explain in the methods.
We have extended this point adding more details of the filter process (lines 170-173).
- L214-215: 4135+2197=6332. This is not 5930, why?
Thanks for noting us this mistake. We have correct it in the text.
- L219: Respectively to …?
We have clarified this point (lines 291-294).
- L221-223: Comparing fig. 4d and fig. 2, a perfect correlation is not possible. Please, check these results.
As detailed above, we have finally discarded our analysis of correlation. Alternatively, we have visually assessed population curves expected per species based on r. We have clarified that point in the Materials and Methods (lines 183-186) as well as in Results (lines 212-214, lines 257-258, Figure 2).
- L223-25: As suggested above, I would remove any analysis with the incomplete year 2021.
We have updated our database, so all analyses now include 2021 as full year.
- L234: Remove “a increase of 500% between 2013 and 2021”.
Done.
- L247-248: You cannot claim this, as you do not have info on the population between 1992 and 2012.
We have added the information available on population estimations before our study in the Materials and Methods, showing that population sizes were much smaller at that time (lines 109-112).
- L264-265: Please, give updated population numbers (e.g., check the new Catalan atlas). On the other hand, I think that monk parakeets arrived to Madrid and Barcelona many years before than rose-ringed parakeets. Monk parakeets have larger populations there simply because they have had more years to thrive.
To our knowledge, there are not synchronic updates for both species in Madrid after censuses performed in 2015 (Molina et al. 2016; Del Moral et al. 2017), so we discard Madrid and have only updated parakeet population sizes of Barcelona and Málaga (Senar et al. 2017; Postigo and Senar 2017). Regarding differences between population sizes of both parakeet species, it is true that an earlier introduction of monk parakeets than rose-ringed parakeets can be one of the main factors to explain this difference, and we have included this possibility in the new version of the manuscript (lines 340-359).
- L274: There are alternative hypotheses to understand the current situation of parakeets in Seville. 1) You know nothing on what happened between 1992 and 2012 (20 years!). Perhaps, the monk parakeets released in 1992 went extinct few years later and the current population was reintroduced later. 2) The monk parakeet is subjected to pest control by removing their nests. Perhaps, these actions constrained population growth in early stages. 3) Finally, as I explained above, the numbers provided for both species are not fully comparable, as they come from the maximum and minimum annual population size.
We understand the comment of the reviewer and we cannot discard them. However, data collected before our study period (presence and estimations of population sizes) of both parakeet species suggest that extinctions during early years after introduction are very unlikely. Regarding the potential effects of nest removal on monk parakeet numbers, we do not think that the little effort done in the study area would be enough to affect population growth. However, we have included this possibility in the new version of the paper (lines 348-350). Finally, it is true that we cannot directly compare our estimates of monk parakeet population sizes with counts of rose-ringed parakeets, not only because they have been done in different moments of their biological cycle but also because of methodological differences. However, having in mind those constraints and warning that absolute numbers are not directly comparable, the fact is that rose-ringed parakeets are much more abundant than monk parakeets. We have explained that in the new version of the ms.
- L293-294: I think the argument should be in the opposite direction. Your data supports citizen science data, not the reverse. However, see my main comments.
We also agree and we have rephrased the sentence (lines 380-385).
- L298: I agree that both species are easily detected in the field, however there are other potential issues overlooked by the authors. 1) Misidentification of both species (plausible for less experienced birders). 2) As they are exotic species, some birders may decide to avoid reporting them. 3) Sampling effort was not homogeneous in your study area. The probability of reporting them in the less crowded cells of your study area is smaller. Thus spatial spread could be representing observers’ presence rather than the true pattern of species expansion.
We have included a brief discussion about these potential biases in the new version of the manuscript (lines 385-407).
- L303-346: This is a nice discussion, but I think it is far from the objectives, data and results of this study. However, I did not see any discussion about the spatial spread of the species.
We have shortened this paragraph to better reflect our study.
- L348-350: I would remove the first sentence of the conclusions.
Done.
Round 2
Reviewer 2 Report
The authors have made a good review job of their manuscript. I really appreciate that they included most of my suggestions. However, I have still some comments. Most of them are minor, but I have some suggestions for the population analyses and the uncertainty of monk parakeet estimations. My comments in order of appearance are:
- L102-5: I would focus only in the period 1992-2012. For instance, the 2015 rose-ringed parakeet census (1367) is used in this study, as part of the authors’ data. So, how can it be a personal communication? Similarly, the 96 monk parakeets of 2015 are quite confusing, as you are providing a value of 199 for that year. How is this possible? Please, check these numbers to keep coherence in your manuscript. Furthermore, in L104, year is incomplete (200?).
- L136: As I pointed out in my previous review, this 1.52 is a key number. Domenech et al. provided a SD=1.8 (range=0-8) for this mean. As you can see in the Domenech et al. paper, they provided n +/- an error. I would expect the same for your numbers of monk parakeets. You may easily estimate a 95% confidence interval for the numbers of monk parakeets by +/- 2*SD/sqrt(n) (being n=129 and SD=1.8). I realize that this is based on a Gaussian distribution of the variable (number of individuals/chamber), which was probably not true (obvious from the SD and range). Nevertheless, an uncertainty of estimated values is always welcome and I strongly suggest you to provide your monk parakeet population size along with a +/- error.
- L149: I would remove “and downloaded”.
- L163: Perhaps, here it is a better place to explain that occurrence data were only a 6%.
- L174: Delete “(“.
- L174: Polynomial regression is perfectly okay, but if you suspect non-lineal temporal trends, GAM would be a preferable choice. In this particular case, as parameter estimates for the polynomial are to some extent irrelevant, a GAM looks even more suitable.
- 190-1: Delete: “representing a population in-190 crease of 425% in nine years”.
- L192: Here, I think there is a mistake. the average should be 0.228 and the range 0.065-0.381. Note that you did not have data for 2014. Thus, the estimated r for 2015 must be divided by 2 as N0 is 2013 and consequently t=2.
In any case, now, I fully understand your approach thanks to your new explanations provided in the methods. Thanks. Honestly, I think that your approach is a bit bizarre. I think that a GLM with a Poisson or NB distribution with year as explanatory variable would be a more straightforward approach. Perhaps, temporal autocorrelation is an issue and it should be checked. The GLM would provide you with true confidence intervals for your exponential growth curve. Furthermore, you could estimate the actual model fitting to your data (e.g. % of explained deviance), instead by a visual inspection, which looks totally arbitrary.
- L211: I suggest “abandoned for unknown reasons”.
- L212: Fig3B. Maybe, the fitted polynomial curve (or GAM) along with their confidence intervals should be shown too.
- L224-5: I think that the p value of the whole model (which I estimated ~0.0014) would be more informative. The model R2 (I estimated ~0.84) would be also really necessary. Alternatively, you should provide the estimates and p-values both for the linear and the quadratic terms of your model. You may also show all these detailed results for your model in a supplementary table.
- L226: As I suggested above, these values should be provided with a +/- error.
- L227: See my previous comment about the GLM for L192.
- L228-9: This result is really unexpected. Looking at the grey bars of the fig 3a, I can see a rather exponential growth in numbers. I would expect the same trend for a value that is just 1.52 times these grey bars. Using your data provided in table S1, I have estimated that an exponential model fits with a R2=0.95 to your data. However, a lineal model is even better: R2=0.97; and a quadratic model is able to fit almost perfectly to your data: R2=0.99 (suggesting some non-linearity). These R2 demonstrate that you may find extremely well fitted GLM to your data, i.e. there is an obvious and extremely significant growth in numbers. You may use the AIC of each model to select the best fitting option.
- L244: “The number of annual volunteer…”
- L247: I think that Fig. 5D is Fig. 6B. Please, avoid repetition.
- L248-250: You should indicate which maps belong to each species.
- L263-6: Provide the dispersion/uncertainty related to these mean values (e.g., SD, or range, etc).
- L287: “study area”.
- L356: Could these decrease be an artifact due to a delayed uploading of records by observers? I mean, perhaps, all info for 2021 is not gathered by the platforms yet.
- L390: “considered, if the responsible”
Here, I am also providing some replies to the authors’ letter. You will find my comments in red. I am only including those replied points. For the rest of points, congrats for a good job and thanks for all explanations.
- There was an important difference between rose-ringed and monk parakeet population surveys: rose-ringed parakeets were counted just after the breeding season, when population reaches its annual peak (adults + juveniles); while monk parakeets nests were surveyed just before the breeding season, when population has its annual minimum (only adults who surveyed from the previous breeding season + recruits). (Note: I am assuming Seville populations are closed and do not receive individuals from far populations). Of course, this difference does not matter to estimate temporal trends for each population. However, it may be important when comparing absolute numbers of both species. Moreover, rose-ringed parakeet data is a true census, as you counted all individuals, while the number of monk parakeet data is an estimation from nests. The (unknown) uncertainty related to the latter number is poorly recognized in the paper and both population values are considered equally real.
We perfectly understand this comment and we have extended this point discussing the differences between both approaches to estimate the number of rose-ringed and monk parakeets in our study (lines 327-332).
R: Honestly, I expected more development of this topic on the discussion beyond a simple claim that both measures are non-comparable. This is obvious for any reader. You may develop further simply adding some of my comments from the first revision round. In addition, I did not see uncertainty estimation for monk parakeet abundance.
- Spatial analysis. The 5x5 km cell approach is ok, but I think that the authors should do a more refined approach by using basic spatial analyses, such as minimum convex polygons or kernel densities. As the authors have the exact locations of the observations, they can estimate accurate distribution areas. As urban areas (ie suitable habitat) are not available homogeneously across the study area, the cell approach is quite poor and uninformative. Moreover, as the sampling effort was not equal in all cells, presence/absence in them cannot be compared.
This is the only suggestion where we do not agree with the reviewer. As we recognize in the manuscript, we just attempt to show a first, roughly approach to the range spread of the two parakeet species.
R: First approach is not incompatible with a thorough approach…
In the case of the nests of monk parakeets monitored by our research team (new Figure 4), we think that the use of cells is adequate. We show the exact location of each nest and a measure of density (the number of chambers in each nest with points increasing in size). However, plotting the observations recorded in citizen science platforms (new Figure 5) is more questionable, as these observations are subject to several sources of bias that are difficult to identify and correct (now discussed in lines 391-407).
R: The new fig. 4 is perfectly fine, in spite of the fact that for the 2021 most of the points are too overlapped to see properly them. The new Fig 5 is also ok. They were already right in the previous version. I am not criticizing your graphical output, which is fine, I was suggesting a different analysis for your spatial data.
Moreover, we conservatively considered the presence of one individual for those observations of the presence of the species where the number of individuals was not indicated by the observer (lines 173-175), while actually the observation could correspond to large flocks. Moreover, observations do not correspond to nests but to individuals or flocks that can be foraging or simply flying to cover large distances from nesting/roosting sites to foraging areas. Altogether, the use of more refined approaches would suffer from the same sources of bias. Despite of all, we still think our approach allows us to show a first, although imperfect, picture of the range expansion of the species using citizen science, with the caveats recognized in Discussion.
R: I agree, but as I suggested you may do a less “imperfect” approach.
On the other hand, we are almost ready to initiate a two-year telemetry project using a large number of tagged individuals from each parakeet species. The information obtained will allow us knowing the exact location of each individual several times at day. It will offer us an invaluable and unbiased information on the movements, use of habitats and space of the two species that will be adequate for the finer spatial analyses suggested by the reviewer.
R: These are excellent news. Congrats for your new telemetry project! However, here, we are discussing about the best approach for your spatial analysis. Despite your long and interesting reply, you did not justify why your approach is better than other alternatives.
- Results from citizen science data. They provide a similar pattern to the one provided by your accurate censuses. However, in my opinion, this may be just a nice coincidence. The exponential growth in the number of observations/individuals provided by citizen science data can be showing simply the exponential growth of the observers’ activity in both projects. There are two results supporting my interpretation: 1) Based on your surveys, monk parakeets increased ~5 times more than rose-ringed parakeets (425% vs 2024%). However, based on citizen science, both species increased a similar amount (6787 vs 6995%). Perhaps, this ~7000% is just the % increase of use of both citizen science platforms from 2013 to 2020. 2) The magnitude of the population increase suggested by the citizen science projects would be extremely overestimated. For instance, compare 425 vs 6787 (16-fold!). In sum, if you do not account for the sampling effort, the total number of records is a meaningless measure. As the sampling effort can be hardly known in non-standardized monitoring schemes, as ebird or observation, you may use the total number of records per year as a proxy for the effort. Alternatively, the % of complete lists reporting the species may be a preferable approach. However, more complex approaches based on state-space models to account for imperfect detection would be the right choice for your analysis.
We thank very much the reviewer for this comments, as we have not clarified in the text that the relationship between the population growth of parakeets and the number of observations in citizen science platforms was not the result of an increment in sampling effort. Thus, in the new version of the ms, we have included linear models instead of correlations where we incorporated a surrogate of sampling effort (i.e., number of observers per year) as a controlling variable. We have detailed this point in Materials and Methods (lines 187-194) as well as in Results (lines 212-214, lines 255-257, new Table 1).
R: Good. However, I have still two comments: 1) The number of observers does not seem the best surrogate for sampling effort. You are assuming that all observers made the same effort among them and among years. If you had all available records (L156), why did not you use the total number of records, as I suggested? 2) As you should be using GLM with a Poisson distribution in your models, the number of observers should be included as an offset variable instead as a covariate. In fact, you are not interested in estimating the “observers” effect. Honestly, from the authors’ explanation, I am concerned about the correctness of the statistical analysis. Perhaps, if they provide their R script, the reviewers could assess their analysis.
Minor comments:
- L2: Title. I suggest to shorten it. For instance, I would remove: “Annual censuses and citizen science data show”. In general, the authors put too much emphasis on the methods and data sources. I am aware that “citizen science” is currently a very fashionable topic, but I would not do an excessive marketing of these key words. Focus on the relevant ecological objectives and findings of your study: invasive parakeets and their population growth.
We have mixing feelings about this minor comment. We are aware our population monitoring results are the strongest part of the paper, while results from citizen science are complementary and –unavoidably- more methodologically questioned. Therefore, it seems reasonable to remove citizen science from the title. However, on the other hand, citizen science has been increasingly used to model distributions and population sizes for a variety of organisms, while lacking real data from these populations to contrast/validate the information derived from citizen science. In this sense, we show here that trends obtained from citizen science are -at least roughly- supported by accurate monitoring programs. We think this point would attract a wide range of researchers and managers, interested on the use of citizen science, much larger than the number of readers just interested on our study species. In the light of these thoughts, we would prefer to maintain citizen science in the title, but we are open to remove it if the reviewer or editor find it more adequate.
R: It is ok to me. My comment was just a suggestion. Authors have the last decision about style issues.
- L131: This value is very important, but the authors did not validate it in their own population after 9 years of study. This is quite surprising…
We understand the comment of the reviewer, and the truth is that we have some counts made in a sample of nests not representative at all of the whole population. However, given that our field surveys required a great effort to cover the study area (735 km2) every year and our monitoring resources are really limited, we were not able to estimate the occupation rate of our whole study population.
R: Note that Domenech et al. estimated this value from only 129 chambers. I am fully aware about your hard field work, but due to the paramount importance of this value, I strongly encourage them to carry out some field observations in their own population.
- L143: What was a “validated observation”? How were observations validated? Who did it? I know that both citizen science projects have local birders that review all records. I assume you are referring to this validation process. If this is the case, please explain it, including details on the procedures used by these birders during validation.
Effectively, this is the validation process. We have added a detailed explanation about that in lines 165-170.
R: Ok, but your description of the validation process looks a quite arbitrary and subjective task. Of course, this is not a criticism towards you, as this is not your business.
- L175: Remove “representing a population increase of 425% in nine years”. It is a non-intuitive way to show the same result again.
Done.
R: Not done… Check
Author Response
REVIEWER: The authors have made a good review job of their manuscript. I really appreciate that they included most of my suggestions. However, I have still some comments. Most of them are minor, but I have some suggestions for the population analyses and the uncertainty of monk parakeet estimations. My comments in order of appearance are:
AUTHORS: Thank you very much for your contribution to enhance our manuscript, we appreciate very much all the detailed suggestions provided. We have addressed all of them since we feel they are helping us to improve the clarity of our manuscript. Changes can be easily seen in the new version with tracked changes.
REVIEWER: L102-5: I would focus only in the period 1992-2012. For instance, the 2015 rose-ringed parakeet census (1367) is used in this study, as part of the authors’ data. So, how can it be a personal communication? Similarly, the 96 monk parakeets of 2015 are quite confusing, as you are providing a value of 199 for that year. How is this possible? Please, check these numbers to keep coherence in your manuscript. Furthermore, in L104, year is incomplete (200?).
AUTHORS: We understand the confusion of the reviewer. We did not mean that all these references are personal communications, but only the 2011census conducted by P. Edelaar, so we have reworded the sentence to clarify this point (line 106). Regarding coherence between census conducted for 2015, we are not meaning that both censuses conducted by Molina et al. (2016) and Del Moral et al. (2017) are used in this article. It is correct that the 2015 rose-ringed parakeet census (Del Moral et al. 2017) agrees with census shown in this study given that some volunteers also collaborated in our study (see Acknowledgments). However, the 2015 monk parakeet census (Molina et al. 2016) is not used in our study because we covered a larger area of distribution of monk parakeets, thus recording more nests (77 nests) than the census conducted by Molina et al. (28 nests) over the same time window (between March and early April). We have noted this discrepancy in the discussion (lines 412-414). Finally, regarding the incomplete year, thanks for noting us this mistake (year 2000). We have correct it in the text.
REVIEWER: L136: As I pointed out in my previous review, this 1.52 is a key number. Domenech et al. provided a SD=1.8 (range=0-8) for this mean. As you can see in the Domenech et al. paper, they provided n +/- an error. I would expect the same for your numbers of monk parakeets. You may easily estimate a 95% confidence interval for the numbers of monk parakeets by +/- 2*SD/sqrt(n) (being n=129 and SD=1.8). I realize that this is based on a Gaussian distribution of the variable (number of individuals/chamber), which was probably not true (obvious from the SD and range). Nevertheless, an uncertainty of estimated values is always welcome and I strongly suggest you to provide your monk parakeet population size along with a +/- error.
AUTHORS: We also agree on this point, so we have added an error to our population size estimates of monk parakeets in Results (line 267) and new Table S1.
REVIEWER: L149: I would remove “and downloaded”.
AUTHORS: Done.
REVIEWER: L163: Perhaps, here it is a better place to explain that occurrence data were only a 6%.
AUTHORS: We also agree on this point, so we have moved this information from Results to this section of Material and Methods (lines 163-165).
REVIEWER: L174: Delete “(“.
AUTHORS: Thanks for noting us this mistake. We have corrected it in the text.
REVIEWER: L174: Polynomial regression is perfectly okay, but if you suspect non-lineal temporal trends, GAM would be a preferable choice. In this particular case, as parameter estimates for the polynomial are to some extent irrelevant, a GAM looks even more suitable.
AUTHORS: As suggested by the reviewer, we have used generalized additive model (GAM) to overcome the nonlinearity in the mean annual number of chambers per nest. We have detailed this point in Materials and Methods (lines 188-193) as well as in Results (lines 261-264, new Figure 3B, new Table S4).
REVIEWER: 190-1: Delete: “representing a population increase of 425% in nine years”.
AUTHORS: We apologize to the reviewer for this mistake, we forgot this suggestion during the previous review of our manuscript. We have deleted it in the new version of the text.
REVIEWER: L192: Here, I think there is a mistake. the average should be 0.228 and the range 0.065-0.381. Note that you did not have data for 2014. Thus, the estimated r for 2015 must be divided by 2 as N0 is 2013 and consequently t=2.
AUTHORS: Thanks for noting us this mistake, we have corrected it in the text (lines 221-222).
REVIEWER: In any case, now, I fully understand your approach thanks to your new explanations provided in the methods. Thanks. Honestly, I think that your approach is a bit bizarre. I think that a GLM with a Poisson or NB distribution with year as explanatory variable would be a more straightforward approach. Perhaps, temporal autocorrelation is an issue and it should be checked. The GLM would provide you with true confidence intervals for your exponential growth curve. Furthermore, you could estimate the actual model fitting to your data (e.g. % of explained deviance), instead by a visual inspection, which looks totally arbitrary.
AUTHORS: As suggested by the reviewer, we have changed our approach to test the population growth of both parakeet species by using GLM. The rose-ringed parakeet model included year as a significant predictor of its population size (exponential growth), while the monk parakeet model included year but in its quadratic form (logistic growth). We have explained these new analyses in Material and Methods (lines 172-178) and Results (220-222 and 265-270, new Figure 2; new Table S2).
REVIEWER: L211: I suggest “abandoned for unknown reasons”.
AUTHORS: Done.
REVIEWER: L212: Fig3B. Maybe, the fitted polynomial curve (or GAM) along with their confidence intervals should be shown too.
AUTHORS: As previously explained, we have changed the Figure 3B to fit the nonlinearity in the mean annual number of chambers per nest over years.
REVIEWER: L224-5: I think that the p value of the whole model (which I estimated ~0.0014) would be more informative. The model R2 (I estimated ~0.84) would be also really necessary. Alternatively, you should provide the estimates and p-values both for the linear and the quadratic terms of your model. You may also show all these detailed results for your model in a supplementary table.
AUTHORS: As suggested by the reviewer previously, we have used generalized additive model (GAM) to overcome the nonlinearity of the mean annual number of chambers per nest. Therefore, we have changed the parameters showed in Results (lines 260-264) and we have added a supplementary table (new Table S4) that shows detailed information for a linear model and a GAM. All the above information was also provided for the GLM used to model population size over years.
REVIEWER: L226: As I suggested above, these values should be provided with a +/- error.
AUTHORS: We have added an error to these estimated values (Results, line 267 and new Table S1).
REVIEWER: L227: See my previous comment about the GLM for L192.
AUTHORS: As previously explained, we have changed our approach to test the population changes over years (GLM; lines 172-178; new Figure 2B, new Table S2).
REVIEWER: L228-9: This result is really unexpected. Looking at the grey bars of the fig 3a, I can see a rather exponential growth in numbers. I would expect the same trend for a value that is just 1.52 times these grey bars. Using your data provided in table S1, I have estimated that an exponential model fits with a R2=0.95 to your data. However, a lineal model is even better: R2=0.97; and a quadratic model is able to fit almost perfectly to your data: R2=0.99 (suggesting some non-linearity). These R2 demonstrate that you may find extremely well fitted GLM to your data, i.e. there is an obvious and extremely significant growth in numbers. You may use the AIC of each model to select the best fitting option.
AUTHORS: As previously mentioned, we have used GLM to test population changes in both parakeet species over years (see Material and Methods, lines 172-178, and Results, 220-222 and 265-270, new Figure 2B). The best fitting option was assessed using AIC (new Table S2).
REVIEWER: L244: “The number of annual volunteer…”
AUTHORS: Done.
REVIEWER: L247: I think that Fig. 5D is Fig. 6B. Please, avoid repetition.
AUTHORS: We have renumbered and moved the figures in the text, making sure that they are correctly referenced.
REVIEWER: L248-250: You should indicate which maps belong to each species.
AUTHORS: We have redone this figure to show the observations of each species separately (new Figure 6).
REVIEWER: L263-6: Provide the dispersion/uncertainty related to these mean values (e.g., SD, or range, etc).
AUTHORS: We have added the SD associated with each mean values (Results, lines 359-362).
REVIEWER: L287: “study area”.
AUTHORS: Thanks for noting us this mistake. We have corrected it in the text.
REVIEWER: L356: Could these decrease be an artifact due to a delayed uploading of records by observers? I mean, perhaps, all info for 2021 is not gathered by the platforms yet.
AUTHORS: Although some observers could upload their data long after performing observations, we think that other issues such as the under-reporting of these common non-native species can have a greater effect on the decrease in the number of observations of parakeets during the last year. However, we have also added this point in the discussion (lines 517-522).
REVIEWER: L390: “considered, if the responsible”
AUTHORS: Done.
REVIEWER: Here, I am also providing some replies to the authors’ letter. You will find my comments in red. I am only including those replied points. For the rest of points, congrats for a good job and thanks for all explanations.
-There was an important difference between rose-ringed and monk parakeet population surveys: rose-ringed parakeets were counted just after the breeding season, when population reaches its annual peak (adults + juveniles); while monk parakeets nests were surveyed just before the breeding season, when population has its annual minimum (only adults who surveyed from the previous breeding season + recruits). (Note: I am assuming Seville populations are closed and do not receive individuals from far populations). Of course, this difference does not matter to estimate temporal trends for each population. However, it may be important when comparing absolute numbers of both species. Moreover, rose-ringed parakeet data is a true census, as you counted all individuals, while the number of monk parakeet data is an estimation from nests. The (unknown) uncertainty related to the latter number is poorly recognized in the paper and both population values are considered equally real.
We perfectly understand this comment and we have extended this point discussing the differences between both approaches to estimate the number of rose-ringed and monk parakeets in our study (lines 327-332).
REVIEWER: Honestly, I expected more development of this topic on the discussion beyond a simple claim that both measures are non-comparable. This is obvious for any reader. You may develop further simply adding some of my comments from the first revision round. In addition, I did not see uncertainty estimation for monk parakeet abundance.
AUTHORS: As suggested by the reviewer, we have extended the discussion on this point (lines 388-392), restructuring the discussion on census methods (lines 388-414). Moreover, we have added an error for the estimated values of the monk parakeet population (see Results, line 267).
- Spatial analysis. The 5x5 km cell approach is ok, but I think that the authors should do a more refined approach by using basic spatial analyses, such as minimum convex polygons or kernel densities. As the authors have the exact locations of the observations, they can estimate accurate distribution areas. As urban areas (ie suitable habitat) are not available homogeneously across the study area, the cell approach is quite poor and uninformative. Moreover, as the sampling effort was not equal in all cells, presence/absence in them cannot be compared.
This is the only suggestion where we do not agree with the reviewer. As we recognize in the manuscript, we just attempt to show a first, roughly approach to the range spread of the two parakeet species.
REVIEWER: First approach is not incompatible with a thorough approach…
AUTHORS: Following the reviewer’s comment, we have added a new analysis focused on the spatial distribution of parakeet records. We have used GLMs to relate whether the temporal trend in cells (5x5 km) recorded as occupied by parakeets in citizen science data (dependent variable) followed our population censuses and, especially, our surveys of monk parakeet nests (independent variables). In these models, we have included the number of cells with bird observations as an offset to control for potential changes in the survey efforts over the study period (see Material and Methods, lines 200-209), as previously suggested by the reviewer. To our knowledge, this control is not possible to be done when dealing with kernels, so we think that this approach is more appropriate when we have to control for variability in sampling effort. Our results showed that spatial citizen science data followed the increment in population size of both parakeets and the spatial expansion of monk parakeets (Results, lines 353-367, new Table 1, new Table S5, new Figure S2). We think these new analyses are closer to the objectives of our study, as we directly tested the potential of citizen science to facilitate spatial information and spread rate of invasive species (see Discussion, lines 499-508 and 522-526). However, we are open to considerer other analysis if the reviewer or editor find them more adequate.
In the case of the nests of monk parakeets monitored by our research team (new Figure 4), we think that the use of cells is adequate. We show the exact location of each nest and a measure of density (the number of chambers in each nest with points increasing in size). However, plotting the observations recorded in citizen science platforms (new Figure 5) is more questionable, as these observations are subject to several sources of bias that are difficult to identify and correct (now discussed in lines 391-407).
REVIEWER: The new fig. 4 is perfectly fine, in spite of the fact that for the 2021 most of the points are too overlapped to see properly them. The new Fig 5 is also ok. They were already right in the previous version. I am not criticizing your graphical output, which is fine, I was suggesting a different analysis for your spatial data.
AUTHORS: We hope the new approach included in this version fulfills the request of the reviewer.
Moreover, we conservatively considered the presence of one individual for those observations of the presence of the species where the number of individuals was not indicated by the observer (lines 173-175), while actually the observation could correspond to large flocks. Moreover, observations do not correspond to nests but to individuals or flocks that can be foraging or simply flying to cover large distances from nesting/roosting sites to foraging areas. Altogether, the use of more refined approaches would suffer from the same sources of bias. Despite of all, we still think our approach allows us to show a first, although imperfect, picture of the range expansion of the species using citizen science, with the caveats recognized in Discussion.
REVIEWER: I agree, but as I suggested you may do a less “imperfect” approach.
AUTHORS: We agree with the reviewer and hope our new approach fulfills his/her requests.
On the other hand, we are almost ready to initiate a two-year telemetry project using a large number of tagged individuals from each parakeet species. The information obtained will allow us knowing the exact location of each individual several times at day. It will offer us an invaluable and unbiased information on the movements, use of habitats and space of the two species that will be adequate for the finer spatial analyses suggested by the reviewer.
REVIEWER: These are excellent news. Congrats for your new telemetry project! However, here, we are discussing about the best approach for your spatial analysis. Despite your long and interesting reply, you did not justify why your approach is better than other alternatives.
AUTHORS: As responded above, we hope the reviewer finds our new analyses suitable.
- Results from citizen science data. They provide a similar pattern to the one provided by your accurate censuses. However, in my opinion, this may be just a nice coincidence. The exponential growth in the number of observations/individuals provided by citizen science data can be showing simply the exponential growth of the observers’ activity in both projects. There are two results supporting my interpretation: 1) Based on your surveys, monk parakeets increased ~5 times more than rose-ringed parakeets (425% vs 2024%). However, based on citizen science, both species increased a similar amount (6787 vs 6995%). Perhaps, this ~7000% is just the % increase of use of both citizen science platforms from 2013 to 2020. 2) The magnitude of the population increase suggested by the citizen science projects would be extremely overestimated. For instance, compare 425 vs 6787 (16-fold!). In sum, if you do not account for the sampling effort, the total number of records is a meaningless measure. As the sampling effort can be hardly known in non-standardized monitoring schemes, as ebird or observation, you may use the total number of records per year as a proxy for the effort. Alternatively, the % of complete lists reporting the species may be a preferable approach. However, more complex approaches based on state-space models to account for imperfect detection would be the right choice for your analysis.
We thank very much the reviewer for this comments, as we have not clarified in the text that the relationship between the population growth of parakeets and the number of observations in citizen science platforms was not the result of an increment in sampling effort. Thus, in the new version of the ms, we have included linear models instead of correlations where we incorporated a surrogate of sampling effort (i.e., number of observers per year) as a controlling variable. We have detailed this point in Materials and Methods (lines 187-194) as well as in Results (lines 212-214, lines 255-257, new Table 1).
REVIEWER: Good. However, I have still two comments: 1) The number of observers does not seem the best surrogate for sampling effort. You are assuming that all observers made the same effort among them and among years. If you had all available records (L156), why did not you use the total number of records, as I suggested? 2) As you should be using GLM with a Poisson distribution in your models, the number of observers should be included as an offset variable instead as a covariate. In fact, you are not interested in estimating the “observers” effect. Honestly, from the authors’ explanation, I am concerned about the correctness of the statistical analysis. Perhaps, if they provide their R script, the reviewers could assess their analysis.
AUTHORS: As suggested by the reviewer, we have redone the models using the annual number of bird observations recorded in citizen science platforms as an offset to control the sampling effort, using a negative binomial distribution to control for overdispersion in our data. These new changes are shown in Material and Methods (lines 193-199) and Results (lines 285-300, new Table 1, new Table S5). Given that these new results are contrary to our previous analysis, we have redone different parts of the Discussion (lines 492-526), the Conclusions (lines 563-567), the Simple Summary (lines 61-63) and Abstract (lines 73-76). We thank the reviewer very much for her/him advice, which allows us to properly deal with this point.
Minor comments:
- L2: Title. I suggest to shorten it. For instance, I would remove: “Annual censuses and citizen science data show”. In general, the authors put too much emphasis on the methods and data sources. I am aware that “citizen science” is currently a very fashionable topic, but I would not do an excessive marketing of these key words. Focus on the relevant ecological objectives and findings of your study: invasive parakeets and their population growth.
We have mixing feelings about this minor comment. We are aware our population monitoring results are the strongest part of the paper, while results from citizen science are complementary and –unavoidably- more methodologically questioned. Therefore, it seems reasonable to remove citizen science from the title. However, on the other hand, citizen science has been increasingly used to model distributions and population sizes for a variety of organisms, while lacking real data from these populations to contrast/validate the information derived from citizen science. In this sense, we show here that trends obtained from citizen science are -at least roughly- supported by accurate monitoring programs. We think this point would attract a wide range of researchers and managers, interested on the use of citizen science, much larger than the number of readers just interested on our study species. In the light of these thoughts, we would prefer to maintain citizen science in the title, but we are open to remove it if the reviewer or editor find it more adequate.
REVIEWER: It is ok to me. My comment was just a suggestion. Authors have the last decision about style issues.
AUTHORS: Thanks for accepting our decision. We have maintained the same title.
- L131: This value is very important, but the authors did not validate it in their own population after 9 years of study. This is quite surprising…
We understand the comment of the reviewer, and the truth is that we have some counts made in a sample of nests not representative at all of the whole population. However, given that our field surveys required a great effort to cover the study area (735 km2) every year and our monitoring resources are really limited, we were not able to estimate the occupation rate of our whole study population.
REVIEWER: Note that Domenech et al. estimated this value from only 129 chambers. I am fully aware about your hard field work, but due to the paramount importance of this value, I strongly encourage them to carry out some field observations in their own population.
AUTHORS: We agree with the reviewer, but unfortunately we do not have information to provide a reliable estimate of the number of monk parakeets per chamber in our study population up to now. We will do it in the next census for sure.
- L143: What was a “validated observation”? How were observations validated? Who did it? I know that both citizen science projects have local birders that review all records. I assume you are referring to this validation process. If this is the case, please explain it, including details on the procedures used by these birders during validation.
Effectively, this is the validation process. We have added a detailed explanation about that in lines 165-170.
REVIEWER: Ok, but your description of the validation process looks a quite arbitrary and subjective task. Of course, this is not a criticism towards you, as this is not your business.
AUTHORS: We cannot intervene in this process, and a criticism on the validation processes is far of our objectives showed in this study. Thus, we simply describe it so readers can think about their reliability.
- L175: Remove “representing a population increase of 425% in nine years”. It is a non-intuitive way to show the same result again.
Done.
REVIEWER: Not done… Check
AUTHORS: As commented above, we apologize to the reviewer for the mistake, we forgot this suggestion during the review process of our manuscript. We have deleted it in the text.
Round 3
Reviewer 2 Report
The authors made an excellent review work of the ms. In fact, I think that the paper has been enormously improved from the first draft that I received. To me, this ms version is fully satisfactory. I thanks to the authors for their good reception of all my suggestions and for including almost all of them.
I have only a couple of final remarks. In Table S4, I think that formulae are wrong for the QM and GAM models. I think that they should be:
Linear model (LM)
Mean number of chambers ~ Year
Quadratic regression model (QM)
Mean number of chambers ~ Year + Year2
Generalized additive model (GAM)
Mean number of chambers ~ fi(Year)
As all these models are quite simple and are perfectly described in the text, I would even suggest to remove these formulae. Moreover, they are rather R code than true mathematical expressions, and consequently its notation can be confusing.
In Fig S2, there is apparently a mistake in the x-axis title in the top monk-parakeet graph. Just check it.
Author Response
- The authors made an excellent review work of the ms. In fact, I think that the paper has been enormously improved from the first draft that I received. To me, this ms version is fully satisfactory. I thanks to the authors for their good reception of all my suggestions and for including almost all of them.
- Thank you very much for these positive comments and for your huge contribution to enhance our manuscript. We have addressed all your final remarks and the changes can be easily seen in the new version of supplementary figures.
- I have only a couple of final remarks. In Table S4, I think that formulae are wrong for the QM and GAM models. I think that they should be:
Linear model (LM)
Mean number of chambers ~ Year
Quadratic regression model (QM)
Mean number of chambers ~ Year + Year2
Generalized additive model (GAM)
Mean number of chambers ~ fi(Year)
As all these models are quite simple and are perfectly described in the text, I would even suggest to remove these formulae. Moreover, they are rather R code than true mathematical expressions, and consequently its notation can be confusing.
- As suggested by the reviewer, we have removed the formulas and only indicated the dependent variable (Number mean of chambers) and each model type (see new Table S4).
- In Fig S2, there is apparently a mistake in the x-axis title in the top monk-parakeet graph. Just check it.
- Thanks noting us this mistake, the correct x-axis title is "Population count". We have corrected it (see new Figure S2).